# Profiling the most elderly parkinson's disease patients: Does age or disease duration matter?

**Sasivimol Virameteekul[1], Onanong Phokaewvarangkul[1], Roongroj Bhidayasiri**[1,2]*

**1** Faculty of Medicine, Department of Medicine, Chulalongkorn Centre of Excellence for Parkinson's Disease & Related Disorders, Chulalongkorn University and King Chulalongkorn Memorial Hospital, Thai Red Cross Society, Bangkok, Thailand, **2** The Academy of Science, The Royal Society of Thailand, Bangkok, Thailand

* rbh@chulapd.org

**Data Availability Statement:** All relevant data are within the paper and its Supporting Information files.

**Funding:** This study was supported by Senior Research Scholar Grant (RTA6280016) of the

## Abstract

### Background

Despite our ageing populations, elderly patients are underrepresented in clinical research, and ageing research is often separate from that of Parkinson's disease (PD). To our knowledge, no previous study has focused on the most elderly ('old-old', age $\geq$ 85 years) patients with PD to reveal how age directly influences PD clinical progression.

### Objective

We compared the clinical characteristics and pharmacological profiles, including complications of levodopa treatment, disease progression, disabilities, and comorbidities of the old-old with those of comparable younger ('young-old', age 60–75 years) PD patients. In addition, within the old-old group, we compared those with a short disease duration (< 10 years at the time of diagnosis) to those with a long disease duration $\geq$10 years to investigate whether prognosis was related to disease progression or aging.

### Methods

This single-centre, case-control study compared 60 old-old to 92 young-old PD patients, matched for disease duration. Patients in the old-old group were also divided equally (30:30) into two subgroups (short and long disease duration) with the same mean age. We compared the groups based on several clinical measures using a conditional logistic regression.

### Results

By study design, there were no differences between age groups when comparing disease duration, however, the proportion of men decreased with age ($p = 0.002$). At a comparable length of PD duration of 10 years, the old-old PD patients predominantly had significantly greater postural instability and gait disturbance ($p = 0.006$), higher motor scope of the Unified Parkinson's Disease Rating Scale (UPDRS-III, $p<0.0001$), and more advanced Hoehn

Thailand Science Research and Innovation (TSRI) (to RB), International Research Network Grant of the Thailand Research Fund (IRN59W0005) (to RB), and Center of Excellence grant of Chulalongkorn University (GCE 6100930004-1) (to RB).

**Competing interests:** There are no conflicts of interest to declare.

& Yahr (H&Y) stage ($p<0.0001$). The Non-Motor Symptoms Questionnaire (NMSQuest) score was also significantly higher among the old-old ($p<0.0001$) compared to the young-old patients. Moreover, the distribution of NMS also differed between ages, with features of gastrointestinal problems ($p<0.0001$), urinary problems ($p = 0.004$), sleep disturbances and fatigue ($p = 0.032$), and cognitive impairment ($p<0.0001$) significantly more common in the old-old group, whereas sexual problems ($p = 0.012$), depression, and anxiety ($p = 0.032$) were more common in the young-old. No differences were found in visual hallucinations, cerebrovascular disease, and miscellaneous domains. While young-old PD patients received higher levodopa equivalent daily doses ($p<0.0001$) and developed a significant greater rate of dyskinesia ($p = 0.002$), no significant difference was observed in the rate of wearing-off ($p = 0.378$). Old-old patients also had greater disability, as measured by the Schwab and England scale ($p<0.0001$) and had greater milestone frequency specifically for dementia ($p<0.0001$), wheelchair placement ($p<0.0001$), nursing home placement ($p = 0.019$), and hospitalisation in the past 1 year ($p = 0.05$). Neither recurrent falls ($p = 0.443$) nor visual hallucinations ($p = 0.607$) were documented significantly more often in the old-old patients.

## Conclusions

Age and disease duration were independently associated with clinical presentation, course, and progression of PD. Age was the main predictor, but disease duration also had a strong effect, suggesting that factors of the ageing process beyond the disease process itself cause PD in the most elderly to be more severe.

## Introduction

One of the most considerable social transformations of the twenty-first century is the increase in the elderly population [1]. Many countries around the globe are currently facing a rapid growth in the number of aging citizens due to low birth rates, longer life expectancies, and the ageing of baby boomers, especially in developed and developing countries (Fig 1) [2,3].

The age composition of Thailand's population is on par with that of many developed countries; it is ranked as the third most rapidly ageing population in Asia, which now stands at about 13 million, accounting for 20% of the population [6]. This situation has resulted in a rise in chronic and degenerative diseases in countries worldwide. Indeed, in the 2015 global burden of disease, injuries, and risk factors study, neurological disorders were listed as the leading cause of disability globally. Amongst these, Parkinson's disease (PD) has the fastest growing prevalence, disability rate, and mortality rate [7].

According to a recent framework, different older adult populations are classified as 'young-old', 'old', and 'old-old' [8–10]. The 'young-old' are the people in their 60s and early 70s who are active and healthy, the 'old' are the people in their 70s and 80s who have chronic illnesses and are slowing down with some bothersome symptoms, and the 'old-old' are the people aged 85 or older who are often sick, disabled, and perhaps even nearing death [11,12].

PD is the most common neurodegenerative movement disorder, and more than 10 million individuals worldwide are estimated to be living with the disease [13]. It is clear that age is the strongest risk factor for PD, with a nearly exponential increase in incidence in patients between the ages of 55 and 79 years (Fig 1) [14–16]. However, the burden of PD in patients

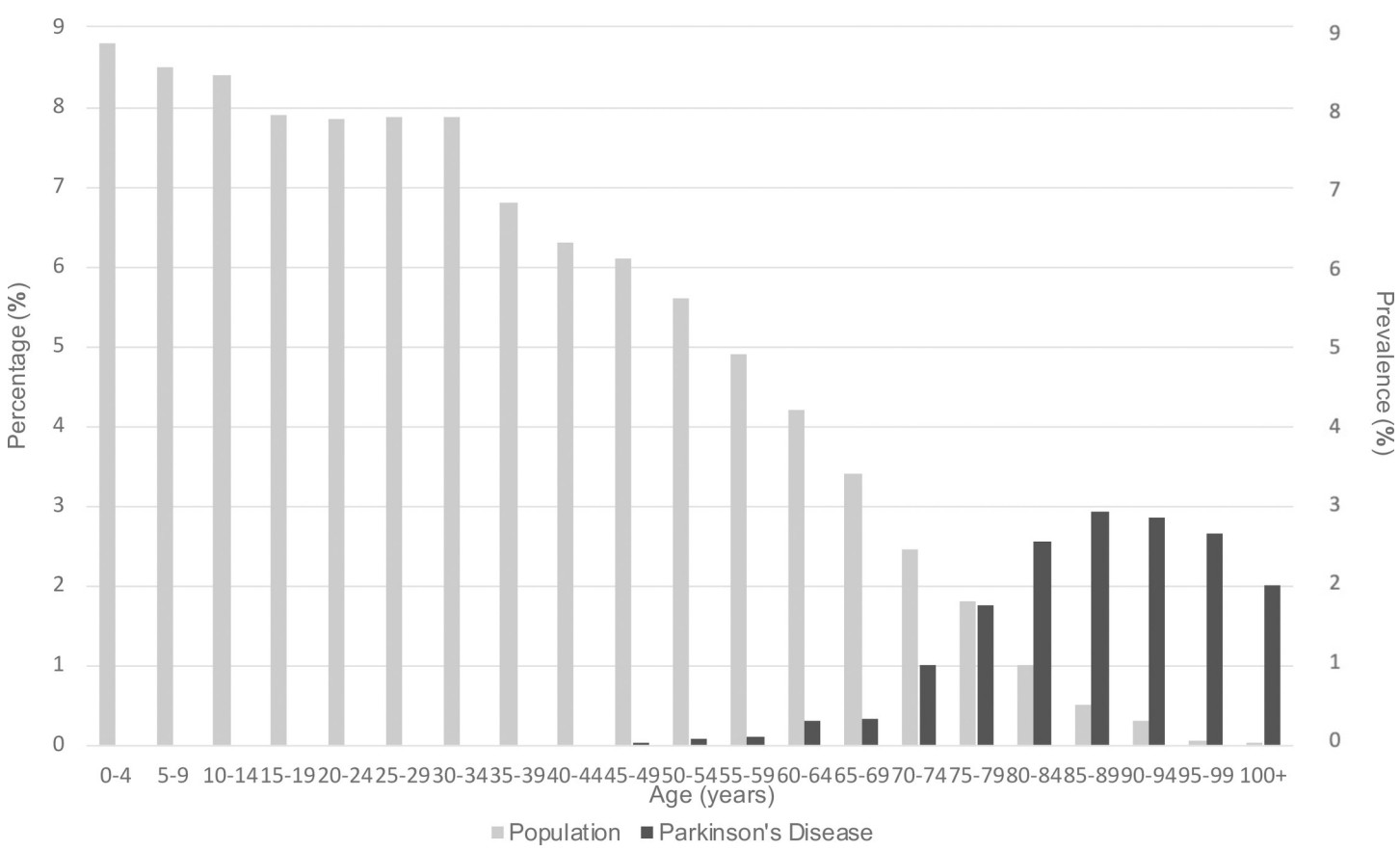

**Fig 1. Distribution of the global population in 2020 [4] and prevalence of global Parkinson's disease in 2016, by age [5].** The total world population amounts to 7,794,798,729. Prevalence is expressed as the percentage of the population that is affected by the disease.

aged 85 years and older remains controversial [14] since they are excluded from the majority of studies [17]. Ageing is a complex phenomenon that affects all the cells of the body, including the dopaminergic neurons of the substantia nigra (SN), and those of the other brain regions specific to PD [18]. Therefore, the accumulation of age-related somatic damage combined with the failure of compensatory mechanisms is important for the manifestation of the clinical characteristics observed in ageing and PD, starting with mild parkinsonian signs that are common in the elderly population, identified in 45% of non-demented healthy elderly individuals [19].

Despite rising patient numbers, elderly patients are underrepresented in clinical research.

Moreover, ageing research is often far removed from that of PD, and it is imperative to bring these two areas together to further our understanding of how age directly influences PD clinical progression.

Data about the most elderly patients with PD is very rare since only a relatively small number of subjects older than 75 years have been included in PD trials. In order to evaluate age bias in PD research, the MEDLINE database was searched from 1999 to 2007, in a previous systemic review [20]. Seventy-nine studies, involving 19,156 patients, were identified for analysis; an estimated 85% of these patients were younger than 75 years, and 94% were younger than 80 years. Older people were excluded from the trials for a variety of reasons [17]. Twenty-three studies (29%) defined an upper age limit (74–86 years) as an exclusion criterion and patients with significant cognitive impairment were excluded from 29 trials (36%). In 12

(15%) and 13 (16%) studies, the presence of psychiatric disturbances and medical comorbidities, respectively, were exclusion criteria [20].

To the best of our knowledge, while several studies have investigated PD patients with late disease onset, few have focused on patients who are at an advanced age, the alone old-old. There is evidence that late-onset PD (LOPD; defined when PD onset is in the late 60s to 70s) patients show more rapid disease progression, greater motor impairment, and poorer response to levodopa compared with that observed in patients with early-onset disease [21–24]. However, data from LOPD patients might not be applicable to all advanced-age PD patients since some of the advanced-age patients have an early disease onset with longer disease duration, and the clinical repercussion of PD in the old-old might be different from that in the old or young-old.

The objective of our study was to compare the clinical characteristics and pharmacological profiles, including complications of levodopa treatment, progression of disease, disabilities, and comorbidities of the old-old with those of comparable young-old patients. Within the old-old group, additional comparison was performed between those with a short disease duration (< 10 years at the time of diagnosis) and those with a long disease duration (≥10 years) to investigate whether the effects observed were related to the disease process or aging.

## Methods

This study was approved by the Human Ethics Committee of the Faculty of Medicine, Chulalongkorn University (IRB. No. 134/62) before study initiation. Written informed consent for the collection of demographics and clinical data was obtained from all study participants. When individuals lacked decision-making capacity, informed consent was obtained from a family member or caregiver.

### Participants

The patients included in this study were evaluated at the Chulalongkorn Centre of Excellence for Parkinson's Disease and Related Disorders (www.chulapd.org) between June 2019 and December 2019. Inclusion criteria were a diagnosis of PD (according to the UK PD Brain Bank criteria) [25] and a record of age at diagnosis. Exclusion criteria included patients who (1) were missing record of exact age, (2) were missing record of the age of diagnosis, or (3) had insufficient clinical data. The patient's actual birth date was also cross-checked against official state documents (such as birth certificate, marriage certificate, identification card, hospital card, and other identity documents), since actual birth dates are often unknown in some elderly individuals in Thailand [26].

Cut-off ages for PD patient age groups were empirically defined, based on the peak representative age ranges in PD and were 85 years or older, called the old-old, and between the ages of 60 and 75 years, called the young-old PD [27–31]. To investigate the influence of disease duration, patients of the old-old and young-old groups were also separated into two subgroups, according to duration of illness, into a 'less than 10 years' group, called the short duration group, and an 'equal to or more than 10 years' group, called the long duration group, based on a commonly used cut-off point from previously published studies [32,33]. Moreover, to address the role of age at onset (AAO), we repeated the analysis in which patients were divided in 2 groups, middle-onset PD (MOPD) group included patients with AAO between 50 and 69, and the LOPD group included patients with AAO ≥70 years. This cut- off was guided by the conclusion of a cluster analysis from Post el al [34]. Based on the available study subjects, we did not study those AAO of less than 50 years.

## Procedures

**Measurement of clinical variables.** Demographic and clinical data extracted included gender, age, age-at-onset, disease duration, levodopa equivalent daily dose (LEDD), presence of motor complications, Hoehn & Yahr (H&Y) stage, annual outpatient visits frequency, and comorbidities. Data instruments included motor section of the Unified Parkinson's Disease Rating Scale (UPDRS-III) [35], rated during the 'on' medication state, as well as the Non-Motor Symptoms Questionnaire (NMSQuest) revised version, which is a self-completed validated questionnaire. Mean NMSQuest scores and frequencies in each domain were corrected to assess patient severity and, thus, to establish the ranking of prevalence for each domain [36]. PD motor subtypes were identified following the original classification methods into two subtypes: (1) tremor- dominant PD and (2) postural instability and gait difficulty (PIGD). Using the UPDRS, an average global tremor score and a mean score for the complex of PIGD were determined, and patients were assigned to a tremor group and a PIGD group based on the ratio of these scores [37,38]. The Charlson Comorbidity Index (CCI) was used to assess comorbidities [39]. The CCI has been shown to be superior to similar scales for predicting mortality in PD patients [40,41].

**Measurement of disease advancement.** Disease advancement was assessed in two ways: (i) the Schwab and England (S&E-ADL) scale, which has been validated for use in PD as a measure of activities of daily living (ADLs, dependency was defined as a score of less than 80%) [42], and (ii) milestones of disease advancement that had been selected on the basis that each was likely to require additional medical attention. Well documented in previous reports [43,44], these were as follows: (1) regular falls separately analysed for ambulatory patients (i.e., H&Y stage 1–4), as a milestone of motor disability, (2) visual hallucinations, (3) dementia, confirmed by applying the Diagnostic and Statistical Manual of Mental Disorders-V criteria (DSM-V) [45], and (4) placement in residential or nursing home care as a measure of global disability. Moreover, other disease milestones, such as hospital or emergency room visits within the past year, and wheelchair dependence were also included.

Information was collected from the patients and cross-checked against medical record charts (patients were followed-up regularly at 3-month intervals on average). When sensory or motor problems interfered with the patient providing information or completing the questionnaire, assistance was provided by close relatives or primary caregivers. All evaluations were carried out by one of the authors (SV) during the 'on' medication state (in the morning, 1–2 hours after taking medications).

## Statistical analysis

Statistical analysis was performed using descriptive statistics. Continuous variables are presented as means and standard deviations (SDs), while categorical variables are presented as frequencies and percentages. Differences in the characteristics of interest were tested using independent samples $t$-test or chi-squared test, where appropriate. Pearson's correlation coefficient was used to evaluate the correlations between the variables and impact measures. Logistic regression was performed using the enter method to identify the variables most likely to predict disease progression. Age and disease duration were used as independent variables, and disease milestones were used as the dependent variables. Significance was set at $p < 0.05$. All statistical analysis was performed using the SPSS software, version 23.0 (SPSS Inc., Chicago IL).

## Results

During the study period, 793 patients were diagnosed with PD in our clinic, of which 65 (8.21%) were classified as old-old PD patients. Five cases were excluded based on the following

reasons: refusal to participate (n = 1), information was unavailable (n = 2), could not be contacted (n = 1), and died before interview (n = 1). After exclusions, the total number of old-old patients was 60. Ninety-two young-old PD cases were selected matched for disease duration from the data file for comparison.

## Old-old and young-old PD cohorts: A general comparison

**Patient characteristics.**  Table 1 shows the demographic and clinical characteristics of all patients with PD in the different age groups. Average age was 88.25 (SD, 3.11 years; range, 85–97 years) for the old-old patients and 66.40 (SD, 3.15 years; range, 60–75 years) for the young-old patients ($p<0.0001$). Average onset age was 77 years (SD, 6.73 years; range, 78–92 years) for the old-old and 56.30 years (SD, 5.33 years; range, 43–66 years) for the young-old group ($p<0.0001$). Disease duration for both the groups was similar (10.50 vs. 10.08, $p = 0.856$). The proportion of male patients decreased with age, with 56 (60.9%) in the young-old group and 24 (40%) in the old-old group ($p<0.001$). The rate of annual outpatient visits in old-old was significantly lower than that of young-old (2.80 vs. 3.29, $p = 0.001$).

**Motor features.**  The old-old group demonstrated a significantly higher frequency of PIGD as the predominant symptom ($p = 0.006$). When comparing the severity of motor symptoms, UPDRS-III scores and H&Y stages were also significantly higher in the old-old PD patients compared to the young-old patients ($p<0.0001$ and $p<0.0001$, respectively). H&Y stage 3 was used as a cut-off to determine postural instability. H&Y stage 3 or above was significantly more frequent among the old-old patients ($p<0.0001$).

**Non-motor features.**  When considering NMS, the mean NMSQuest score was significantly greater amongst the old-old patients than the young-old patients ($p < 0.0001$). The most frequent NMS in the old-old group was gastrointestinal problems (90%), followed by cognitive deficits (81.7%), and urinary problems (73.3%). Gastrointestinal problems (73.9%), sexual problems (57.6%), and sleep disturbances and fatigue (54.3%) were the most common NMS among the young-old patients. Distribution of NMS also differed between the age groups, with features of gastrointestinal problems ($p<0.0001$), urinary problems ($p = 0.004$), sleep disturbances and fatigue ($p = 0.032$), and cognitive impairment ($p<0.0001$) significantly more common in the old-old patients, whereas sexual problems ($p = 0.012$), depression, and anxiety ($p = 0.032$) were more common in the young-old patients. No differences were found in visual hallucinations ($p = 0.955$), cardiovascular issues ($p = 0.098$), or miscellaneous domains ($p = 0.077$).

**Dopaminergic medications.**  All patients in both groups were taking at least one PD medication (levodopa, a dopamine agonist, monoamine oxidase B inhibitor, or entacapone) regardless of age. When calculating the LEDD, the mean dosage in the old-old patients was 555 mg and in the young-old patients it was 890 mg ($p<0.0001$). The old-old group had a significantly lower rate of dyskinesia than the young-old group (16.7% vs. 40.2%, $p = 0.002$). However, wearing-off rate did not differ between the groups ($p = 0.378$).

**Comorbidities.**  Old-old patients were more likely to have comorbid conditions than the young-old patients. The most common specific comorbidities were cerebrovascular disease ($p = 0.005$) and musculoskeletal disease ($p<0.0001$), whereas the prevalence of hypertension ($p = 0.685$), diabetes ($p = 0.512$), and cancer ($p = 0.340$) was not significantly different. Comorbidity severity as measured by CCI was also significantly higher in the old-old patients ($2.2 \pm 1.2$ vs. $1.0 \pm 1.16$, $p<0.0001$).

**Measures of disease advancement.**  Milestone frequency was generally higher in the old-old group. Significant differences were observed in the prevalence of dementia ($p<0.0001$), wheelchair placement ($p<0.0001$), nursing home placement ($p = 0.019$), and hospitalisation in

**Table 1. Comparison of demographic and clinical characteristics of old-old versus young-old PD patients.**

| | Age (yrs) | | p-value |
|---|---|---|---|
| | 60–75 (N = 92) | ≥ 85 (N = 60) | |
| **Demographic variables** | | | |
| Current age, yrs, mean (±SD) | 66.40 (±3.51) | 88.25 (±3.11) | <0.0001* |
| Age of PD onset, yrs, mean (±SD) | 56.30 (±5.33) | 77.73 (±6.73) | <0.0001* |
| Disease duration, yrs, mean (±SD) | 10.08 (±5.26) | 10.50 (±6.63) | 0.856 |
| Gender, male, N (%) | 56 (60.9%) | 24 (40%) | 0.002* |
| Annual outpatient visits, mean (±SD) | 3.29 (±1.00) | 2.80 (±0.75) | 0.001* |
| **Motor symptoms** | | | |
| Predominant subtype, N (%) | | | |
| TD | 40 (43.5%) | 13 (21.7%) | 0.006* |
| PIGD | 52 (56.5%) | 47 (78.3%) | |
| Motor severity | | | |
| UPDRS-III, mean (±SD) | 27.86 (±14.27) | 41.82 (±17.51) | <0.0001* |
| H&Y stage, mean (±SD) | 2.84 (±0.87) | 4.24 (±0.88) | <0.0001* |
| PI-H&Y(H&Y>3), N (%) | 15 (16.3%) | 48 (80%) | <0.0001* |
| **Non motor symptoms, N (%)** | | | |
| NMSQuest total, mean (±SD) | 8.16 (±2.80) | 11.93 (±3.03) | <0.0001* |
| Domain, N (%) | | | |
| Gastrointestinal tract | 68 (73.9%) | 54 (90%) | <0.0001* |
| Urinary tract | 46 (50%) | 44 (73.3%) | 0.004* |
| Sexual function | 53 (57.6%) | 22 (36.7%) | 0.012* |
| Cardiovascular issues | 25 (27.2%) | 24 (40%) | 0.098 |
| Sleep/fatigue | 50 (54.3%) | 71.1 (63.3%) | 0.032* |
| Apathy/attention/memory | 20 (21.7%) | 49 (81.7%) | <0.0001* |
| Hallucination/delusion | 28 (30.4%) | 18 (30%) | 0.955 |
| Depression/anxiety | 42 (45.7%) | 17 (28.3%) | 0.032* |
| Miscellaneous | 48 (52.2%) | 40 (66.7%) | 0.077 |
| TMSE, mean (±SD) | 27.04 (±3.51) | 17.48 (±7.78) | <0.0001* |
| **Medications** | | | |
| LED, mg/d, mean (±SD) | 890.79 (±546.79) | 555 (±336.95) | <0.0001* |
| LED > 400 | 76 (82.6%) | 31 (51.7%) | <0.0001* |
| **Motor complication, N (%)** | | | |
| Dyskinesia | 37 (40.2%) | 10 (16.7%) | 0.002* |
| Wearing-off | 44 (47.8%) | 33 (55%) | 0.378 |
| **Disabilities** | | | |
| S&E-ADL, mean (±SD) | 78.80 (±18.08) | 46.5 (±22.83) | <0.0001* |
| S&E-ADL < 80%, N (%) | 30 (32.6%) | 49 (81.6%) | <0.0001* |
| Milestones, N (%) | | | |
| Dementia | 8 (8.7%) | 34 (56.7%) | <0.0001* |
| Recurrent falls in ambulatory patient (N = 116), N (%) | 21/85 (24.7%) | 8/31 (25.8%) | 0.904 |
| Visual hallucination | 33 (35.9%) | 24 (40%) | 0.607 |
| Nursing home placement | 3 (3.3%) | 8 (13.3%) | 0.019* |
| Wheelchair placement | 11(12%) | 32(53.3%) | <0.0001* |
| Hospitalization in past year, N (%) | 14 (15.2%) | 21 (35%) | 0.05* |
| **Comorbidity, N (%)** | | | |
| CVD | 14 (15.2%) | 21 (35%) | 0.005* |
| Musculoskeletal | 30 (32.6%) | 39 (65%) | <0.001* |

*(Continued)*

**Table 1.** (Continued)

| | Age (yrs) | | p-value |
|---|---|---|---|
| | 60–75 (N = 92) | ≥ 85 (N = 60) | |
| Hypertension | 16 (17.4%) | 12 (20%) | 0.685 |
| Diabetes mellitus | 16 (17.4%) | 13 (21.7%) | 0.512 |
| Cancer | 2 (2.2%) | 3 (5%) | 0.340 |
| CCI, mean (±SD) | 1.0 (±1.16) | 2.2 (±1.2) | <0.0001* |

TD, tremor-dominant; PIGD, postural instability/gait difficulty; UPDRS, Unified Parkinson's Disease Rating Scale; H&Y, Hoehn & Yahr; NMSQuest, Non-Motor Symptoms Questionnaire; TMSE, Thai Mental State Examination; LED, levodopa equivalent dose; S&E-ADL, Schwab and England Activities of Daily Living, CVD, cerebrovascular disease; CCI, Charlson Cormorbidity Index.

the past year ($p = 0.05$). Neither recurrent falls ($p = 0.904$) nor visual hallucinations ($p = 0.607$) were documented significantly more often in the old-old patients. Old-old patients also had greater disability, as measured by the S&E-ADL scale ($p<0.0001$) and tended to have greater dependency, as recorded as a score of less than 80% on the S&E-ADL scale (81.7% vs. 32.6%, $p<0.0001$), compared to young-old patients.

## Old-old PD patients with short and long disease duration: A general comparison

The 60 patients in the old-old groups were separated equally (30:30) into two subgroups, those with short disease duration ($< 10$ years) and those with long disease duration ($\geq 10$ years). Subgroups had the same mean age ($p = 0.266$), but a mean disease duration of 4.87 and 16.13 years, respectively ($p<0.0001$). Gender ratio did not differ significantly ($p = 0.598$).

Table 2 shows the demographic data and baseline clinical characteristics of both subgroups.

The long disease duration group showed higher motor scores on the UPDRS-III scale ($p<0.0001$) and average H&Y staging ($p = 0.002$) than the short disease duration group; however, the frequency of the patients with H&Y stage 3 or higher was similar in both groups ($p = 0.095$). Tremor as a predominant symptom was found less frequently in the longer disease duration group ($p = 0.028$). The rate of annual outpatient visits in long disease duration group (2.60 times per person per annum; pppa) was significantly lower than that of short disease duration (3.00 pppa) ($p = 0.039$).

**Non-motor features.** Patients with longer disease duration reported significantly higher mean NMSQuest scores than those with short disease duration (13.30 [± 2.85] vs. 10.57 [± 2.64], $p<0.0001$). More frequently reported problems by the patients with longer PD duration on the NMSQuest were in the domain of urinary problems ($p = 0.004$), followed by hallucinations and delusions ($p = 0.024$). There were no statistically significant differences in the remaining domains (gastrointestinal problems, $p = 0.389$; sexual problems, $p = 0.592$; cerebrovascular disease, $p = 0.292$, cognitive deficits, $p = 0.317$; sleep disturbances/fatigue, $p = 0.774$; depression/anxiety, $p = 0.152$; and miscellaneous, $p = 0.573$).

In addition, there was no statistically significant difference between the two groups with respect to LEDD ($p = 0.127$). Wearing-off was the most frequently reported motor complication in both groups, and, as expected, was associated with longer disease duration ($p = 0.004$). Dyskinesia was much less frequent, and there was no significant difference between the two groups ($p = 0.083$).

**Comorbidities.** There were no differences in comorbidity frequency or comorbidity severity (CCI, $p = 0.672$) between the short and long disease duration groups, except for musculoskeletal disease ($p = 0.015$).

**Table 2. Comparison of demographic and clinical characteristics for old-old PD patients with disease duration <10 years versus those with disease duration ≥10 years.**

| | Disease duration | | p-value |
|---|---|---|---|
| | <10 (N = 30) | ≥ 10 (N = 30) | |
| **Demographic variables** | | | |
| Current age, yrs, mean (±SD) | 87.80 (±3.01) | 88.70 (±3.196) | 0.266 |
| Age of PD onset, yrs, mean (±SD) | 82.93 (±3.0) | 72.53 (±5.21) | <0.0001* |
| Disease duration, yrs, mean (±SD) | 4.87 (2.69) | 16.13 (±4.08) | <0.0001* |
| Gender, male, N (%) | 11 (36.7%) | 13 (43.3%) | 0.598 |
| Annual outpatient visits, mean (±SD) | 3.00 (±0.78) | 2.60 (±0.67) | 0.039 * |
| **Motor symptoms** | | | |
| Predominant subtype, N (%) | | | |
| TD | 10 (33.3%) | 3 (10%) | 0.028 |
| PIGD | 20 (66.7%) | 27 (90%) | |
| Motor severity | | | |
| UPDRS-III, mean (±SD) | 30.13 (±13.19) | 53.5 (±12.94) | <0.0001* |
| H&Y, mean (±SD) | 3.9 (±0.89) | 4.58 (±0.72) | 0.002* |
| Pi-H&Y(H&Y>3), N (%) | 22 (73.3%) | 27 (90%) | 0.095 |
| **Non-motor symptoms, N (%)** | | | |
| NMSQuest total, mean (±SD) | 10.57 (±2.64) | 13.30 (±2.85) | <0.0001* |
| Domain, N (%) | | | |
| Gastrointestinal | 26 (86.7%) | 28 (93.3%) | 0.389 |
| Urinary | 17 (56.7%) | 27 (90%) | 0.004* |
| Sexual | 10 (33.3%) | 12 (40%) | 0.592 |
| CVS | 10 (3.3%) | 14 (46.7%) | 0.292 |
| Sleep/fatigue | 21 (70%) | 22 (73.3%) | 0.774 |
| Apathy/attention/memory | 23 (76.7%) | 26 (86.7%) | 0.317 |
| Hallucination/delusion | 5 (16.7%) | 13 (43.3%) | 0.024* |
| Depression/anxiety | 6 (20%) | 11 (36.7%) | 0.152 |
| Miscellaneous | 20 (66.7%) | 22 (73.3%) | 0.573 |
| TMSE, mean (±SD) | 18.97(±6.98) | 16 (±8.35) | 0.141 |
| **Medications** | | | |
| LEDD, mg/d, mean (±SD) | 488.33(±267) | 621.67(±387) | 0.127 |
| **Motor complication, N (%)** | | | |
| Dyskinesia | 2 (6.7%) | 8 (26.7%) | 0.083 |
| Wearing-off | 11 (36.7%) | 22 (73.3%) | 0.004* |
| **Disabilities** | | | |
| S&E-ADL, mean (±SD) | 55 (±22.1) | 38 (±20.57) | 0.003* |
| S&E-ADL ≥ 80%, N (%) | 8 (26.7%) | 3 (10%) | 0.095 |
| Milestones, N (%) | | | |
| Dementia | 15 (50%) | 19 (63.3%) | 0.297 |
| Recurrent falls in ambulatory patient (N = 31), N (%) | 6/22 (27.3%) | 2/9 (22.2%) | 0.771 |
| Visual hallucination | 8 (26.7%) | 16 (53.3%) | 0.035* |
| Nursing home placement | 3(10%) | 5(16.7%) | 0.448 |
| Wheelchair placement | 12(40%) | 20(66.7%) | 0.038* |
| Hospitalization in past year, N (%) | 9 (30%) | 12 (40%) | 0.417 |
| **Comorbidity, N (%)** | | | |
| CVD | 9 (30%) | 12 (40%) | 0.417 |
| Musculoskeletal | 15 (50%) | 24 (80%) | 0.015* |

*(Continued)*

**Table 2.** (Continued)

| | Disease duration | | *p*-value |
|---|---|---|---|
| | <10 (N = 30) | ≥ 10 (N = 30) | |
| Hypertension | 5 (16.7%) | 7 (23.3%) | 0.519 |
| Diabetes | 7 (23.3%) | 6 (20%) | 0.754 |
| Cancer | 3 (10%) | 0 | 0.076 |
| CCI, mean (±SD) | 2.13 (±1.22) | 2.26 (±1.20) | 0.672 |

TD, tremor-dominant; PIGD, postural instability/gait difficulty; UPDRS, Unified Parkinson's Disease Rating Scale; H&Y-S, Hoehn & Yahr staging; NMSQuest, Non-Motor Symptoms Questionnaire; TMSE, Thai Mental State Examination; LED, levodopa equivalent dose; S&E-ADL, Schwab and England Activities of Daily Living, CVD, cerebrovascular disease; CCI, Charlson Cormorbidity Index.

## Measurement of disease advancement

Concerns about disease milestones, visual hallucinations, and wheelchair-dependence were significantly more frequent in the long disease duration group ($p$ = 0.035 and $p$ = 0.038, respectively). Other advanced disease milestones were also more frequently reported in the long disease duration group, although these differences were not statistically significant.

## MOPD and LOPD: A general comparison

S1 Table shows details of demographic and clinical characteristics of the PD regarding to AAO. 97 patients (63%) belonged in the MOPD group and 55 (37%) in the LOPD group. Current age was 67.63 ±6.05 and 88.07 ±3.97 years in the MOPD and LOPD group, respectively ($p$<0.0001), whereas disease duration for both the groups was similar (10.93 vs. 9.04, $p$ = 0.054). Tremor or PIGD as the predominant motor symptom was not significantly different in MOPD and LOPD ($p$ = 0.544). UPDRS-III scores and H&Y stages were significantly higher in the LOPD patients compared to the MOPD patients ($p$<0.0001 and $p$<0.0001, respectively).

Considering NMS, the mean NMSQuest score was significantly greater amongst the LOPD patients than the MOPD patients ($p$<0.0001). More frequently reported NMS by MOPD patients were in the domain of sexual function ($p$ = 0.016), followed by depression and anxiety ($p$ = 0.024). While features of gastrointestinal problems ($p$<0.0001) and cognitive impairment ($p$<0.0001) were significantly more reported in the LOPD patients. There were no statistically significant differences in the remaining domains.

When calculating the LEDD, the mean dosage in the MOPD patients was 559 mg and in the LOPD patients it was 871 mg ($p$<0.0001). Dyskinesia developed in 40% of the MOPD and 14.5% of the LOPD patients (p<0.001), while wearing-off rate did not differ between the groups ($p$ = 0.701).

For disabilities and important clinical milestones, LOPD patients were more likely to have a greater disability, as measured by the S&E-ADL scale ($p$<0.0001) and tended to have higher milestone frequency. Specific milestones more common in LOPD than in MOPD patients were dementia ($p$<0.0001), wheelchair placement ($p$<0.0001), and hospitalisation in the past year ($p$ = 0.011).

Our study extended these comparisons to the young-old patients with short and long disease duration. However, as to the available subjects, there was a significant different in age at examination (S2 Table).

## Correlation and regression analyses

Correlation outcomes are shown in S3 Table. Pearson's correlation revealed that age and disease duration were strongly correlated with several variables. In particular, there was a

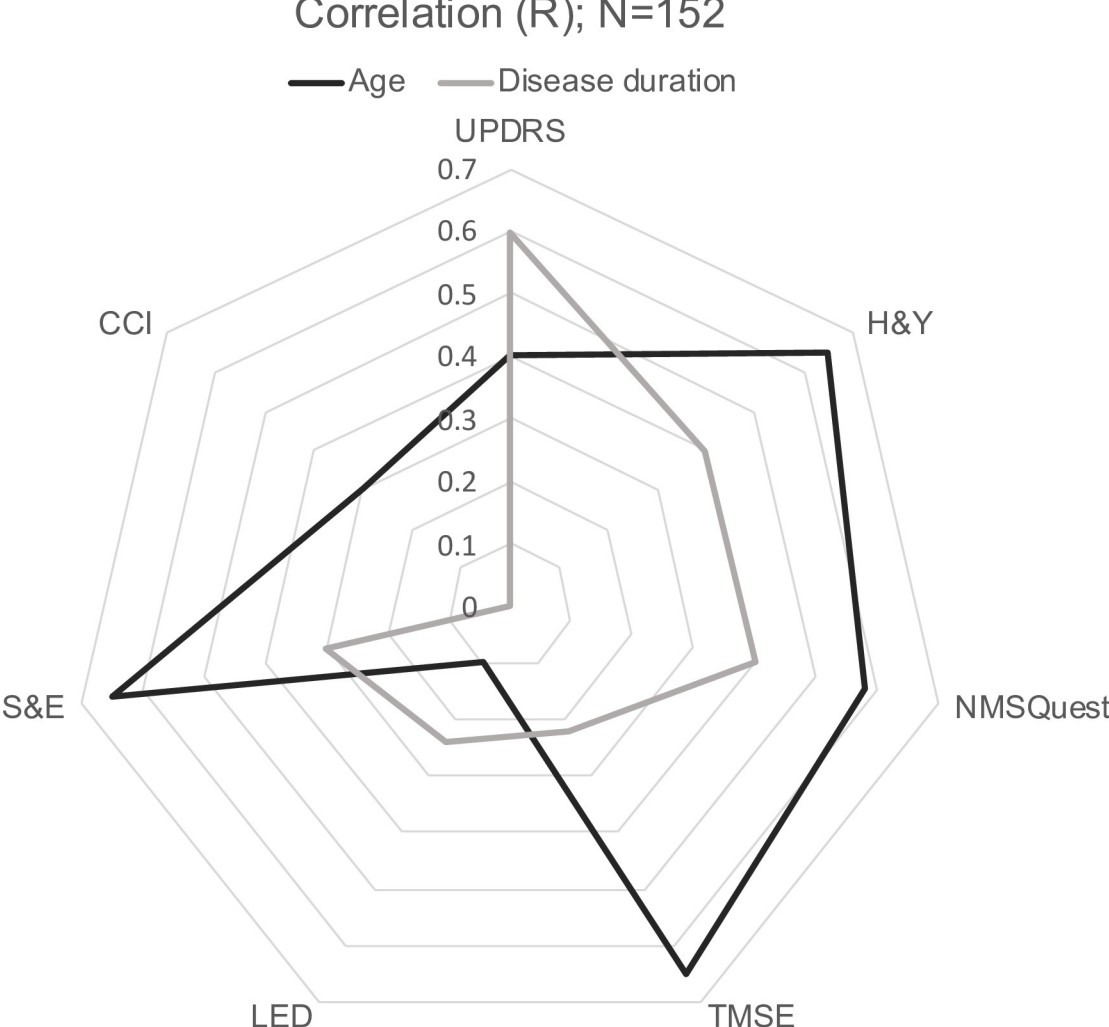

**Fig 2. Radar graph of absolute correlations between age, disease duration and disease variables.** UPDRS, Unified Parkinson's Disease Rating Scale; H&Y, Hoehn & Yahr; NMSQuest, Non-Motor Symptoms Questionnaire; TMSE, Thai Mental State Examination; LED, levodopa equivalent dose; S&E-ADL, Schwab and England Activities of Daily Living; CCI, Charlson Cormorbidity Index.

correlation between older age and higher UPDRS-III score (r = 0.471, $p < 0.0001$), more advanced H&Y stage (r = 0.657, $p < 0.0001$), higher NMSQuest score (r = 0.553, $p < 0.0001$), lower Thai Mental State Examination (TMSE) score (r = -0.665, $p < 0.0001$), lower LEDD (r = -0.283, $p < 0.0001$), lower S&E-ADL score (r = -0.666, $p < 0.0001$), and higher CCI score (r = 0.466, $p < 0.0001$). Disease duration was also correlated with greater UPDRS-III score (r = 0.595, $p < 0.0001$), more advanced H&Y stage (r = 0.402, $p = 0.015$), higher NMSQuest score (r = 0.399, $p < 0.0001$), lower TMSE score (r = -0.236, $p = 0.003$), higher LEDD (r = 0.250, $p = 0.002$), and lower S&E-ADL score (r = -0.451, $p < 0.001$). Advance disease duration showed no significant correlation with CCI score. Fig 2 shows a different perspective on the relationship between age, disease duration, and each disease variable.

A logistic regression model was created to predict the presence of advanced disease milestones using independent variables, including age and disease duration, as detailed in S4 Table. For specific disease milestones, a highly significant model resulted where the old-old

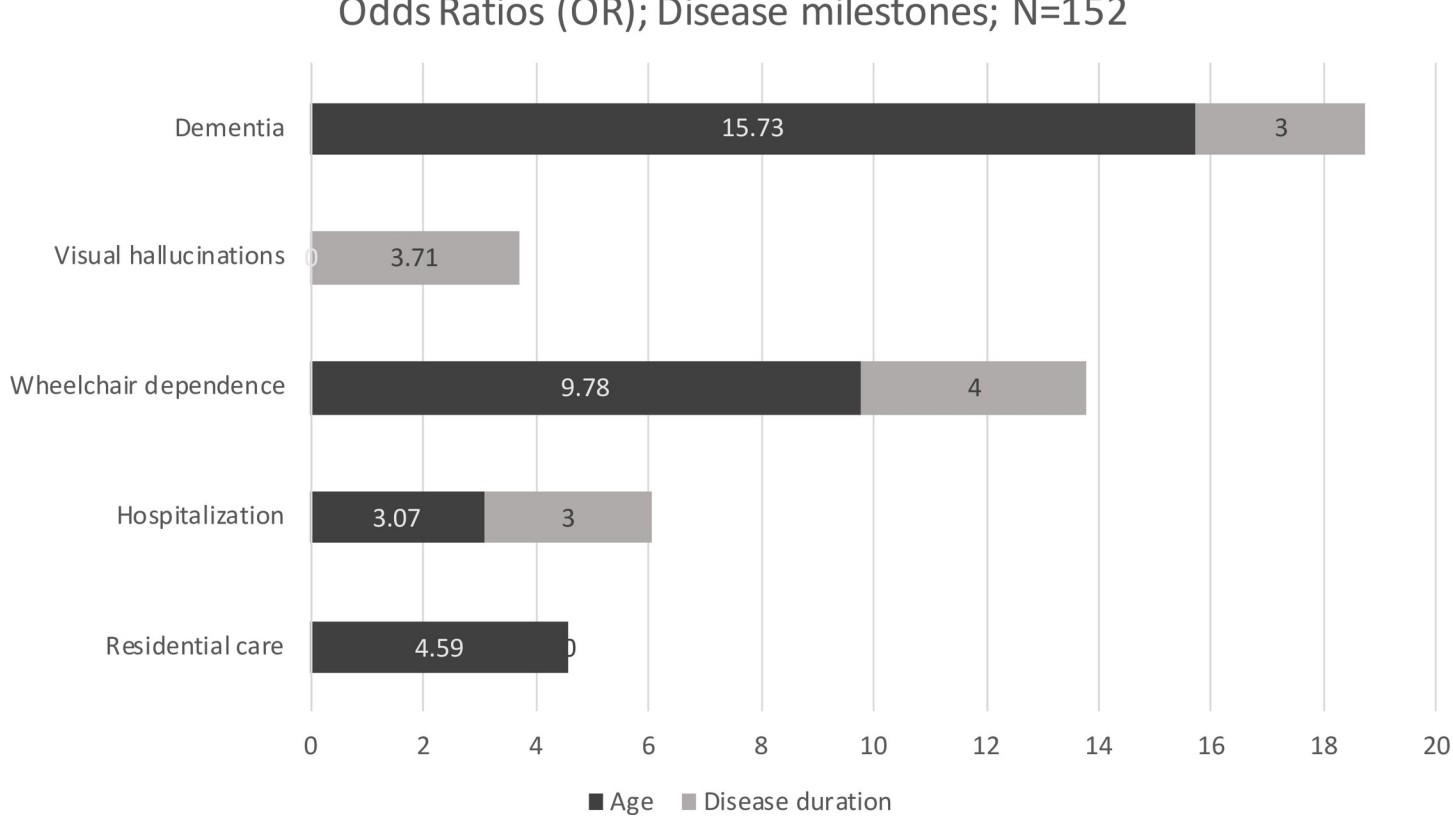

**Fig 3. Radar graph of odds ratio visualization for disease milestones difference by age and disease duration.**

PD patients were 15.7 times more likely to develop dementia, 9.8 times more likely to be wheelchair dependent, 3.1 times more likely to be hospitalised in the past year, and 4.6 times more likely to be in residential care than the young-old patients. Also, PD patients with 10-year disease duration or greater were 3.2 times more likely to develop dementia, 3.7 times more likely to be wheelchair dependence, 2.5 times more likely to be hospitalised in the past year, and 3.7 times more likely to have visual hallucinations than the young-old patients. Fig 3 shows the odds ratio visualisation for differences in disease milestones by age and disease duration.

## Discussion

To better represent old-old patients, this study compared the profile of patients aged 85 years or older to those of patients in age groups with the highest prevalence rate (60–75 years). Unlike previous studies, we aimed to relate the role of ageing to clinical progression of the disease rather than to clinical onset or disease initiation, and to examine the role of disease duration in subgroup analysis. Overall, compared with young-old PD patients, old-old patients had more severe PD motor and non-motor phenotypes, global disability, higher prevalence of motor complications, and heavier comorbidity burden. Another important finding is that some of these specific features are even greater with longer disease duration, and not just in advanced age groups, but also in other age groups too.

In the old-old group, the proportion of male patients was significantly lower (male:female ratio 1:1.5) compared to that of the young-old group and the reported overall PD ratio (1.5–

2.5 times higher prevalence in men [46–49]). As in the normal elderly as well as in patients with other chronic diseases, male life expectancy has been reported to be lower than female life expectancy [50,51]. Female gender may be a protective factor for longer life expectancy, possibly related to hormonal or other gender-specific factors [46,52]. Moreover, this difference might reflect differences in the pathophysiology of PD in younger versus older individuals, wherein environmental factors might play a weaker role in elderly men and oestrogen no longer has a protective role in elderly women.

The PIGD/Tremor predominant (TD) ratio increased with older age and longer disease duration. Previous studies support our findings that the PIGD-subtype propensity is influenced by age and medical comorbidities [53,54]. The common thread here is that older age and more medical comorbidities disproportionately give rise to more axial motor impairments that lead to postural instability and gait difficulty as predominant features [55]. Additionally, longitudinal studies have also shown that with disease progression, TD may transition to the PIGD subtype, suggesting that the motor subtype classification is not stable over time [56,57]. Moreover, there is evidence that PIGD is strongly related to the development of cognitive decline, worsening parkinsonian motor burden, and leading to greater levodopa resistance [53,58]. These results indicate PIGD phenotype to be, not a discrete subtype, but rather a new stage in the progression of PD from early brainstem-localized pathology towards a more widespread multisystem brain disorder influenced by several overlapping age-related pathologies.

At the same disease duration, of approximately 10 years, the old-old patients had greater motor impairment, as rated by the UPDRS-III scale and H&Y stage, compared to the young-old patients. NMS assessed by the NMSQuest total score were also greater in the old-old patients. The mean score in the old-old PD patients was 11.92 (severe severity [36,59,60]), while the mean score in the young-old patients was 8.16 (moderate severity). We found that all patients had at least one NMS. The difference was evident for gastrointestinal, urinary, cognitive, and sleep and fatigue domains. We postulated that the convergence of deficits in multiple transmitter systems and pathways, including the cholinergic, noradrenergic, and serotonergic systems, which are also common in the ageing brain, may all be associated with the clinical expression of NMS. Gjerloff and colleagues investigated the parasympathetic involvement of acetylcholinesterase binding using 11C-donepezil positron emission tomography and reported early enteric cholinergic dysfunction in PD [61,62]. Dominant noradrenergic deficit has also been proposed to be the key to dysautonomia in PD [63,64]. Furthermore, studies have demonstrated that sleep dysfunction in PD is associated with reduced serotonergic function in the midbrain raphe, basal ganglia, and hypothalamus [65,66] and the pathophysiology of cognitive impairment in PD may be cortical dominant with cholinergic dysfunction [62,67–69]. Within the old-old group, patients with longer disease duration had worse NMSQuest score than those with a short disease duration; however, the frequency in each domain was not significantly higher with duration except for urinary problems and visual hallucinations/delusion. It is known that a range of NMS, most notably impaired sense of smell, sleep dysfunction, and dysautonomia are present in PD from the prodromal phase [70,71] to the final palliative stage [71].

As hypothesised, less aggressive treatment strategies were used in the old-old patients. We recognise that these treatment strategies reflect only one centre's approach, but we suggest that they reflect a more generalised reluctance to use complex treatment and higher dosage in the old-old patients. Complex interactions between comorbidity, polypharmacy, altered pharmacodynamics, and pharmacokinetics means there is much merit in this 'start low and go slow' approach [72–74]. In the same way, old-old patients had lower mean annual medical visits than the young-old age group. This might be because individuals aged 85 years and over, particularly those with longer disease duration, are frailer. This limits their ability to travel to

clinic, or make them more likely to require either inpatient care or nursing home replacement where medical care is usually provided.

In the present study, the prevalence of wearing-off did not increase with age, but increased with increasing disease duration and disease severity, whereas dyskinesia was decreased in the old-old patients. The wearing-off rate for both the old-old (55%) and young-old (47.8%) groups in our study was similar to that reported in the DATATOP study [75], which reported high rates of wearing-off and dyskinesia in approximately 50% and 30% of patients after 2 years of levodopa treatment, respectively. The reported prevalence of motor complications in PD patients shows a wide range, from 3% to 94%, mostly 30%–74% [76–79], and it is generally believed that they usually develop in approximately 50% of patients on treatment with levodopa after 5 years [80,81]. Compared with those reported in other Asian populations, a Japanese cross-sectional multicentre study reported that the incidence of wearing-off was 21.3%, 59.4%, and 73.2% at the end of the 5th, 10th, and 15th year after disease onset [78]; in a Chinese multicentre registry survey, the incidence of wearing-off and dyskinesia was 46.5% and 10.3%, respectively [82]. In the present study, the rate of dyskinesia in the old-old patients (16.7%) was both lower than that in the young-old patients (40.2%) and that previously reported, which might be explained by lower levodopa dosage and perhaps reflects this group poorer capacity to exhibit maladaptive plastic responses. The STRIDE-PD analysis also supported this finding, showing that young age was associated with an increased risk of dyskinesia; however, since the age limit in the STRIDE-PD study was 30–70 years, our study further expands these findings to old-old patients [83].

We also assessed disabilities and important clinical milestones, which have been previously proposed to carry important prognostic information, and appear valuable as markers of neuropathological disease stage [43,44]. We explored whether advanced age or advanced disease duration contributed to a more advanced stage of the disease in the old-old patients. Based on logistic analysis, we found that age and disease duration were both independently associated with the occurrence of disease milestones. Age was a stronger predictor of dementia and wheelchair dependence, while advanced disease duration was a stronger predictor of visual hallucinations. The regular fall rate was surprisingly low in patients aged > 85 years. Since the rate of wheelchair dependence is very high in this age group, it is plausible that most of them were not ambulatory patients. In contrast, in the young-old group, who were mostly independently mobile, balance was often impaired, and falls were frequent. Therefore, we separately analysed the fall rate in ambulatory patients (H&Y stage 1–4), however, no significant differences were found. Another unexpected finding was that the prevalence of patients living at home (87%) was high. This finding contrasts with the Sydney multicentre study that 25% of all PD patients were admitted to a nursing home within 10 years of diagnosis [84], and Auyeung's report that 27% were institutionalised within 10 years [85]. Previous studies have shown that residential care placement was related not only to the factors determining health status, such as age, comorbidity, and dependence on personal ADLs, but that living conditions, marital status, financial status, education, and culture also exert a significant influence [86]. In Thai culture, the elderly prefer to remain at home with support from formal caregivers and family members, which could explain the high prevalence of our patients living at home [87,88]. Our observation that there is a strong correlation between disease duration and visual hallucination concurs with that of Kempster et al., who found that the relationship between the clinical milestones and Lewy body pathology was strongest for the dementia and visual hallucinations milestones, while physical disability was not strongly correlated with cortical Lewy body deposition, and the link between residential care and falling milestones was weaker [43]. Other studies have suggested that α-synuclein, tau, and amyloid-β deposition in the limbic regions may have an additive effect in causing cognitive impairment in the elderly [89,90]. Four

milestones, visual hallucinations, recurrent falls, dementia, and nursing home placement, have emerged as markers of advanced disease stage. Our observations have some practical applications in that wheelchair dependence could be useful as a clinical milestone for mobility disability in patients over 85 years of age, and hospitalisation for medical decisions rather than institutional placement could be considered as a milestone in the Thai elderly populations.

It is generally considered that ageing is the biggest risk factor for developing PD; however, not all individuals, including those over 80 years of age, are affected by PD with advancing age, and global prevalence of PD is only 1% to 2%. The mechanisms of aging and PD are complex and interrelated, sharing important biological features, including impairment of the neuronal repair system, mitochondrial dysfunction, neuronal protein aggregation, and metal toxicity, which resulting in increased levels of reactive oxygen species leading to cellular damage [15,91,92]. However, unlike ageing, PD involves factors or mechanisms that produces a regionally specific and more selective dopaminergic neuron loss in the SN [93]. Our study demonstrates that when PD is superimposed on the very old brain, the outcome is a more severe clinical expression and disability profile than that seen in younger patients. These distinctive phenotypes of PD in old-old patients are related to numerous age-related factors including the overlap between signs of senescence and parkinsonism, confounding comorbidities resulting in higher motor impairment, less aggressive treatment, and loss of therapeutic efficacy due to age-related alterations in pharmacokinetics and pharmacodynamics. A final explanation for the observed differences could be related to the different natural courses of pathological processes and cellular pathways in very elderly PD patients. First, younger PD cases show a clear typical pattern of Lewy body (LB) pathology, as predicted by Braak staging [94]. In these cases, there is a slow pathological progression of LB pathology that relates to slow clinical progression, which is consistent with the timing of the appearance of dementia in PD. In contrast, old-old PD patients have very high diffuse loads of LBs that either occur at the onset of clinical disease or rapidly infiltrate the brain. This greater plaque pathology, along with the overlapping AD pathology could support a more aggressive phenotype and rapid clinical progression of PD to a dementia syndrome [95]. Second, studies have shown that dopaminergic neuronal populations seem preferentially vulnerable to loss with ageing compared to the neuronal populations in many other brain regions including the regions related to other neurodegenerative disorders, such as the hippocampus [96–98]. Furthermore, the subregional pattern of striatal dopamine loss in normal ageing differs substantially from the pattern typically observed in PD [93,99]. The ventral SN is more severely affected in PD, while the dorsal subdivision is more severely affected in normal aging. When considering parkinsonian signs associated with the degeneration of the dorsal SN, another study reported that stooped posture, postural instability, and body bradykinesia, which were common amongst the old-old PD in our study, were independently associated with lower neuron density in this subdivision [100]. Also, the prevalence of non-dopaminergic lesions in elderly individuals has been increasingly reported.

This study aims to fill the gap in the literature concerning the profile of old-old Parkinson's patients and to demonstrate whether the driving force in each characteristic is worsening of the disease pathophysiology with increasing duration or age-related processes of the advanced age. To address the role of AAO, we repeated our analysis using AAO of PD as a comparator, detailed in S1 and S2 Tables. In general, outcomes were similar to our other findings, as an inevitable consequence of variable-dependence around age, AAO and disease duration. Previous studies have suggested that AAO may contribute to the distinctive clinical-biochemical-pathological profile of patients with AAO $\leq$ 50 years (i.e., young-, early-, juvenile-onset PD; YOPD), while the different profile seen in patients with AAO > 50 years (middle- and late-onset PD) may be based on the make-up of the very old brain, including, rate of nigrostriatal

degeneration, reduced compensatory mechanisms, and frequency of comorbidities [101,102]. To reflect this, and to avoid confusion due to different pathophysiology bases of YOPD, our studies specifically focused on those with AAO >50 years.

## Limitations

There are some limitations to this study that require further discussion. First, our study was a single hospital-based study with a relatively small sample size. Since the old-old PD patients account for approximately 8.2% of PD cases, our results cannot be generalised to all old-old PD patients in the population, although we included all cases in our unit and collected data on all disease dimensions. As young-old patients were match selected to control the confounding effect of disease duration, enrolment could not be applied to the entire cohort. This could potentially create a bias in the study subjects, however proper matching was accomplished. Second, the common features of older age, such as a larger number of comorbidities and greater intake of medications compared to the younger individuals might lead to higher scores on clinical scales, particularly for NMS. Moreover, obtaining accurate data from the old-old can be more challenging, as cognitive, hearing, and visual impairments act as barriers. Therefore, we advocate the use of simplified explanations and protocols, fewer exclusion criteria, and easier physical access for the old-old PD patients in future studies. Another potential limitation is the restricted longitudinal data collection to six years due to restricted hospital electronic health records. The temporal relationship variables and disease outcomes, thus, can be limited. We strongly suggest that longitudinal studies would enable a robust overall evidence base in a future research.

## Conclusions

While age is known to be the strongest risk factor of PD, old-old patients have often been excluded from PD studies. This study fills this knowledge gap and evaluates the differences in the disease and social factors between the old-old and young-old PD patients. We hope our findings have acknowledged the differential contribution of ageing and disease progression to various disease and social dimensions in old-old patients with PD and provide the rationale for better understanding and targeted management for this population of PD.

## Supporting information

**S1 Table. Comparison of demographic and clinical characteristics of middle-onset PD versus late-onset PD patient.** TD, tremor-dominant; PIGD, postural instability/gait difficulty; UPDRS, Unified Parkinson's Disease Rating Scale; H&Y-S, Hoehn & Yahr staging; NMSQuest, Non-Motor Symptoms Questionnaire; TMSE, Thai Mental State Examination; LED, levodopa equivalent dose; S&E-ADL, Schwab and England Activities of Daily Living, CVD, cerebrovascular disease; CCI, Charlson Cormorbidity Index.
(DOCX)

**S2 Table. Comparison of demographic and clinical characteristics for younger-old PD patients with those disease duration <10 years versus those ≥10 years.** TD, tremor-dominant; PIGD, postural instability/gait difficulty; UPDRS, Unified Parkinson's Disease Rating Scale; H&Y-S, Hoehn & Yahr staging; NMSQuest, Non-Motor Symptoms Questionnaire; TMSE, Thai Mental State Examination; LED, levodopa equivalent dose; S&E-ADL, Schwab and England Activities of Daily Living, CVD, cerebrovascular disease; CCI, Charlson Cormorbidity Index.
(DOCX)

**S3 Table. Correlation of various variables with age and disease duration.** UPDRS, Unified Parkinson's Disease Rating Scale; H&Y, Hoehn & Yahr; NMSQuest, Non-Motor Symptoms Questionnaire; TMSE, Thai Mental State Examination; LED, levodopa equivalent dose; S&E-ADL, Schwab and England Activities of Daily Living, CCI, Charlson Cormorbidity Index. [a] Lower scores indicate greater disability.
(DOCX)

**S4 Table. Logistic regression model for disabilities and milestones.** Model summary. a Hosmer and Lemeshow test 0.403, Nagelkerke R square 0.099. b Hosmer and Lemeshow test 0.782, Nagelkerke R square 0.338. c Hosmer and Lemeshow test 0.406, Nagelkerke R square 0.126. d Hosmer and Lemeshow test 0.449, Nagelkerke R square 0.129. e Hosmer and Lemeshow test 0.034, Nagelkerke R square 0.044. f Hosmer and Lemeshow test 0.972, Nagelkerke R square 0.125.
(DOCX)

## Author Contributions

**Conceptualization:** Roongroj Bhidayasiri.

**Data curation:** Sasivimol Virameteekul, Onanong Phokaewvarangkul.

**Formal analysis:** Sasivimol Virameteekul, Onanong Phokaewvarangkul.

**Funding acquisition:** Roongroj Bhidayasiri.

**Investigation:** Roongroj Bhidayasiri.

**Methodology:** Sasivimol Virameteekul, Onanong Phokaewvarangkul, Roongroj Bhidayasiri.

**Project administration:** Roongroj Bhidayasiri.

**Resources:** Roongroj Bhidayasiri.

**Supervision:** Roongroj Bhidayasiri.

**Validation:** Sasivimol Virameteekul, Onanong Phokaewvarangkul.

**Visualization:** Sasivimol Virameteekul, Roongroj Bhidayasiri.

**Writing – original draft:** Sasivimol Virameteekul.

**Writing – review & editing:** Roongroj Bhidayasiri.

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
