## [Decision Letter · Decision Letter 0]

9 Jun 2021

PONE-D-21-03548

Profiling the Most Elderly Parkinson’s Disease Patients: Does Age or Disease Duration Matter?

PLOS ONE

Dear Dr. Bhidayasiri,

Thank you for submitting your manuscript to PLOS ONE. After careful consideration, we feel that it has merit but does not fully meet PLOS ONE’s publication criteria as it currently stands. Therefore, we invite you to submit a revised version of the manuscript that addresses the points raised during the review process.

We look forward to receiving your revised manuscript.

Kind regards,

Karsten Witt

Academic Editor

PLOS ONE

Journal Requirements:

Reviewers' comments:

Reviewer's Responses to Questions

**Comments to the Author**

1. Is the manuscript technically sound, and do the data support the conclusions?

Reviewer #1: Yes

Reviewer #2: Yes

2. Has the statistical analysis been performed appropriately and rigorously? 

Reviewer #1: Yes

Reviewer #2: Yes

3. Have the authors made all data underlying the findings in their manuscript fully available?

Reviewer #1: No

Reviewer #2: Yes

4. Is the manuscript presented in an intelligible fashion and written in standard English?

Reviewer #1: Yes

Reviewer #2: Yes

5. Review Comments to the Author

Reviewer #1: Virameteekul et al present a well conducted descriptive case control study comparing the characteristics of Parkinson’s disease in older patients, a group that is commonly not included in other studies of PD. These data are important and very interesting. Their analyses to separate the effects of age and disease duration are well presented.

Major comments:

Why was a random subset of young-old PD patients selected, rather than including the entire cohort? Their analyses could be conducted (with more power) with the entire group if that data is available.

The abstract states that the young-old patients were selected to be matched for disease duration; the manuscript reports that a random selection was made: this should be clarified.

In the discussion, the authors present what seems to be a systematic review of the literature and report the age of patients in prior studies. This analysis is interesting and would be more appropriate in the results. They also state that “the mean age in these studies was significantly lower than …” -- the authors should report the statistics.

The authors could exclude or separately analyze H&Y stage 5 patients in their assessment of fall rate, as these patients should be nonambulatory (as noted in their discussion).

The authors note in their methods that visits were conducted approximately every 3 months. It would be interesting to present summary statistics about the visit frequency in the results - was there a difference between the groups? Was there a relationship between visit frequency and age or disease severity, as older or more advanced patients may find it more difficult to travel to a clinic?

Minor:

Minor spelling error: “Demographic” in table 1

In the abstract and at end of the introduction (line 124), the long disease duration group should be noted as “≥10” rather than just “10”.

Figure 2: Is the absolute value of R being plotted (as R is negative for some variables)? If so this should be specified.

Reviewer #2: This is a cross-sectional study of Parkinson’s disease (PD) that concentrates on the disease characteristics of ‘old-old’ (≥ 85 years) patients. The authors remark that the very elderly have been under-represented in previous PD research.

Comments and criticisms:

1. While there is little that is new or unexpected in the findings of this study, the authors argue for some originality in their focus on the old-old age group.

2. The manuscript is quite long. INTRODUCTION begins with a paragraph on ageing in general. It would be better just to concentrate on ageing in relation to PD.

3. This is a study of PD in old-old age, rather than of older onset PD. There are patients here with PD onset in middle age but long disease courses. The short and long disease duration comparison for the old-old group helps to highlight this.

4. Both young-old and old-old patients are survivors of cohorts that were diagnosed with PD on average 10 years before. Can information be given about the size and outcome of the original cohorts? How many have died, or been lost to follow up? For old-old, this involves accounting for non-survivors who would have passed 85 years of age had they still been attending the clinic.

5. I can’t see criteria or references for subtyping of PD into postural instability-gait disturbance and tremor-dominant.

6. There is a tendency for readers to try to draw inferences about the longitudinal character of PD from cross-sectional or retrospective studies. This applies to quite a lot of PD research from the past. Point 4 above refers to one source of potential distortion. LIMITATIONS should discuss this issue.

6. PLOS authors have the option to publish the peer review history of their article (what does this mean?). If published, this will include your full peer review and any attached files.

Reviewer #1: No

Reviewer #2: No

---

## [Author Response · Author response to Decision Letter 0]

4 Jul 2021

Please see uploaded response letter if the format of tables is misplaced.

04 July 2021

Dr. Karsten Witt

Academic Editor

PLOS ONE

Dear Dr. Witt,

Re: Manuscript # PONE-D-21-03548: Profiling the Most Elderly Parkinson’s Disease Patients: Does Age or Disease Duration Matter?

We are grateful to the editor and reviewers for their time and constructive comments on our manuscript. We have implemented their comments and suggestions and wish to submit a revised version of the manuscript for further consideration in the journal. Changes in the manuscript are highlighted in red in the revised version. Below, we also provide a point-by-point response explaining how we have addressed each of the editors or reviewers’ comments.

As reviewer #1 comment on our manuscript regarding data policy. We have addressed this issue in the statistical analysis section (page 8, line 191-192), where it read: “Data were collected in Excel files, encrypted, anonymised, and stored on ChulaPD’s secure data server for analysis”

Reviewers' comments:

Reviewer #1: Virameteekul et al present a well conducted descriptive case control study comparing the characteristics of Parkinson’s disease in older patients, a group that is commonly not included in other studies of PD. These data are important and very interesting. Their analyses to separate the effects of age and disease duration are well presented.

Response: We appreciate the reviewer’s interest and constructive comments of our manuscript. 

Major comments:

1. Why was a random subset of young-old PD patients selected, rather than including the entire cohort? Their analyses could be conducted (with more power) with the entire group if that data is available.

Response: 

Thank you for raising this point. In this study, young-old patients were selected to match for disease duration in order to analyse the influence of age (without confounding factor from disease duration). Moreover, some were excluded according to the exclusion criteria, therefore, we resorted to analysing data from 92 cases age between 60 and 75 years. 

To define this issue, we included the following statement in the limitation section (page27, line523-525). “Furthermore, as the young-old patients were selected to control the confounding effect of disease duration, enrolment could not be applied to the entire cohort. Therefore, a bias in the study subjects could potentially have been created.” 

2. The abstract states that the young-old patients were selected to be matched for disease duration; the manuscript reports that a random selection was made: this should be clarified.

Response: 

Thank you for bringing this inconsistency to our attention. To address this concern, we have replaced the wording of “randomly selected” in results of the manuscript with “matched for disease duration” (see page 9, line 209-210), where it read, “Ninety-two young-old PD cases were selected, matched for disease duration from the data file for comparison” 

Indeed in this sentence, as mention in point 1, we meant to say “ we matched for disease duration to accomplish our research question”. a limitation. (page27, line523-525)

“Furthermore, as the young-old patients were selected to control the confounding effect of disease duration, enrolment could not be applied to the entire cohort. Therefore, a bias in the study subjects could potentially have been created.”

3. In the discussion, the authors present what seems to be a systematic review of the literature and report the age of patients in prior studies. This analysis is interesting and would be more appropriate in the results. They also state that “the mean age in these studies was significantly lower than …” -- the authors should report the statistics.

Response: 

This systemic review was actually conducted by Rajapakse A et al. to demonstrate the present of age bias in clinical trials of Parkinson’s disease. We agree with you that this work is really interesting, therefore we have included it in our manuscript. This is why we could not bring it in our result. However, to ensure the flow of the manuscript, we moved this part to the introduction and further clarified by adding “a previous systemic review” (page5, line105) and also inserted the reference. (reference number 20, see page5, line112 of the introduction of the manuscript). 

20. Rajapakse A, Rajapakse S, Playfer J. Age bias in clinical trials of Parkinson's disease treatment. J Am Geriatr Soc. 2008;56(12):2353-4.

We also omitted the sentence “The mean age in these studies was significantly lower than the age groups with the highest prevalence and incidence rates of PD given by published epidemiological studies from 1999 to 2007.”, as this might cause misunderstanding and the previous review did not provide any information regarding the statistics. To conclude, this paragraph has been rewritten as following (see page5, line103-112).

“Data about the most elderly patients with PD is very rare since only a relatively small number of subjects older than 75 years have been included in PD trials. In order to evaluate age bias in PD research, a previous systemic review searched the MEDLINE database for PD trials from 1999 to 2007 [20]. Seventy-nine studies, involving 19,156 patients, were identified for analysis; an estimated 85% of these patients were younger than 75 years, and 94% were younger than 80 years. Older people were excluded from the trials for a variety of reasons [15]. Twenty-three studies (29%) defined an upper age limit (74–86 years) as an exclusion criterion and patients with significant cognitive impairment were excluded from 29 trials (36%). In 12 (15%) and 13 (16%) studies, the presence of psychiatric disturbances and medical comorbidities, respectively, were exclusion criteria [20].”

4. The authors could exclude or separately analyze H&Y stage 5 patients in their assessment of fall rate, as these patients should be non-ambulatory (as noted in their discussion).

Response: As suggested by the reviewer, we have re-analysed the fall rate separately for ambulatory patients (i.e., H&Y stage 1-4). Then we have modified the results in the table 1 and 2 as below (see the Disabilities sub-section of Table1; page12 and Table 2; page17) and also revised the text to emphasize this point as following “(1) regular falls separately analysed for ambulatory patient (i.e., H&Y stage 1-4), as a milestone of motor disability” (see page8, line179 in the method of the manuscript), “Neither recurrent falls (p=0.904) nor visual hallucinations (p=0.607) were documented significantly more often in the old-old patients.” (page14, line 267 in the results), and “Therefore, we separately analysed the fall rate in ambulatory patients only (H&Y stage 1-4), however, no significant differences were found.” (page24, line 459-460 in the discussion section)

Table 1: Comparison of demographic and clinical characteristics of old-old versus younger-old PD patient (Disabilities sub-section) 

Disabilities 

S&E-ADL, mean (�SD) 78.80 (�18.08) 46.5 (�22.83) <0.0001*

S&E-ADL < 80%, N (%) 30 (32.6%) 49 (81.6%) <0.0001*

Milestones, N (%) 

Dementia 8 (8.7%) 34 (56.7%) <0.0001*

Recurrent falls in ambulatory patient (N=116), N (%) 21/ 85 (24.7%) 8/ 31 (25.8%) 0.904

Visual hallucination 33 (35.9%) 24 (40%) 0.607

Nursing home placement 3 (3.3%) 8 (13.3%) 0.019*

Wheelchair placement 11(12%) 32(53.3%) <0.0001*

Hospitalization in past year, N(%) 14 (15.2%) 21 (35%) 0.05*

Table 2: Comparison of demographic and clinical characteristics for old-old PD patients with those disease duration <10 years versus those ≥10 years (Disabilities sub-section)

Disabilities 

S&E-ADL, mean (�SD) 55 (�22.1) 38 (�20.57) 0.003*

S&E-ADL ≥ 80%, N (%) 8 (26.7%) 3 (10%) 0.095

Milestones, N (%) 

Dementia 15 (50%) 19 (63.3%) 0.297

Recurrent falls in ambulatory patient (N=31), N (%) 6/ 22 (27.3%) 2/ 9(22.2%) 0.771

Visual hallucination 8 (26.7%) 16 (53.3%) 0.035*

Nursing home placement 3(10%) 5(16.7%) 0.448

Wheelchair placement 12(40%) 20(66.7%) 0.038*

Hospitalization in past year, N (%) 9 (30%) 12 (40%) 0.417

5. The authors note in their methods that visits were conducted approximately every 3 months. It would be interesting to present summary statistics about the visit frequency in the results - was there a difference between the groups? Was there a relationship between visit frequency and age or disease severity, as older or more advanced patients may find it more difficult to travel to a clinic?

Response: We welcome the opportunity of presenting the data on number of outpatient visits in our manuscript. As requested, this information has been added in the Table 1, 2, and text (Results section- page10, line 220-221, and page15, line285-287, as well as Discussion section- page 23, line 421-425) where it read, 

Table 1: Comparison of demographic and clinical characteristics of old-old versus younger-old PD patient

 Age (yrs) p-value

 60-75 (N = 92) ≥ 85 (N = 60) 

Demographic variables 

Current age, yrs, mean (�SD) 66.40 (�3.51) 88.25 (�3.11) <0.0001*

Age of PD onset, yrs, mean (�SD) 56.30 (�5.33) 77.73 (�6.73) <0.0001*

Disease duration, yrs, mean (�SD) 10.08 (�5.26) 10.50 (�6.63) 0.856

Gender, male, N (%) 56 (60.9%) 24 (40%) 0.002*

Annual outpatient visits, 

mean (�SD) 3.29 (�1.00) 2.80 (�0.75) 0.001*

Table 2: Comparison of demographic and clinical characteristics for old-old PD patients with those disease duration <10 years versus those ≥10 years 

 Disease duration p-value

 <10 (N = 30) ≥ 10 (N = 30) 

Demographic variables 

Current age, yrs, mean (�SD) 87.80 (�3.01) 88.70 (�3.196) 0.266

Age of PD onset, yrs, mean (�SD) 82.93 (�3.0) 72.53 (�5.21) <0.0001*

Disease duration, yrs, mean (�SD) 4.87 (2.69) 16.13 (�4.08) <0.0001*

Gender, male, N (%) 11 (36.7%) 13 (43.3%) 0.598

Annual outpatient visits, mean (�SD) 3.00 (�0.78) 2.60 (�0.67) 0.039 *

Results section- page 10, line 220-221

“The rate of annual outpatient visits in old-old was significantly lower than that of young-old (2.80 vs. 3.29, p=0.001).”

Page 15, line 285-287

“The rate of outpatient visits in long disease duration group (2.60 times per person per annum; pppa) was significantly lower than that of short disease duration (3.00 pppa).” 

Discussion section- page 23, line 421-425

“In the same way, old-old patient had fewer mean annual medical visits than the young-old age group. This might be because individuals aged 85 years and over, particularly those with longer disease duration, are frailer. This limits their ability to travel to clinic, or make them more likely to require either inpatient care or nursing home placement where medical care is usually provided in situ.”

Minor comments:

6. Minor spelling error: “Demographic” in table 1

Response: We have corrected this spelling error

7. In the abstract and at end of the introduction (line 124), the long disease duration group should be noted as “≥10” rather than just “10”.

Response: Thank you for pointing this out. We have made this change according to your suggestion. 

8. Figure 2: Is the absolute value of R being plotted (as R is negative for some variables)? If so this should be specified.

Response: Thank you for pointing this out. We have made this change according to your suggestion

Reviewer #2: This is a cross-sectional study of Parkinson’s disease (PD) that concentrates on the disease characteristics of ‘old-old’ (≥ 85 years) patients. The authors remark that the very elderly have been under-represented in previous PD research.

1. While there is little that is new or unexpected in the findings of this study, the authors argue for some originality in their focus on the old-old age group.

Response: Thank you for giving us the opportunity to address this concern. While the reviewer is correct that the findings may not be unexpected, they have not been systematically conducted in the literature as evident by a systemic review (reference number 20). Moreover, late-onset PD patients as reported in the literature (reference number 21-24) are not identical to advanced-age PD patients as some of the advanced-age patients may have an early disease onset with longer disease duration. We are hoping that our study has provided a number of new findings to better understand different factors that potentially contribute to the outcomes of old-old PD patients. The following information has been included in the introduction to address this concern (page 5, line 114- 125):

“To the best of our knowledge, while several studies have investigated PD patients with late disease onset, few have focused on patients who are at an advanced age, not to mention the old-old. There is evidence that late-onset PD (LOPD; defined when PD onset is in the late 60s to 70s) patients show more rapid disease progression, greater motor impairment, and poorer response to levodopa compared with that observed in patients with early-onset disease [21-24]. However, data from LOPD patients might not be applicable to all advanced-age PD patients since some of the advanced-age patients have an early disease onset with longer disease duration, and the clinical repercussion of PD in the old-old might be different from that in the old or young-old. Therefore, it seems that elderly PD patients are underrepresented in clinical research. Moreover, ageing research is often far removed from that of PD, and it is imperative to bring these two areas together to further our understanding of how age directly influences PD clinical progression.”

2. The manuscript is quite long. INTRODUCTION begins with a paragraph on ageing in general. It would be better just to concentrate on ageing in relation to PD.

Response: As suggested by the reviewer, we have shortened the introduction and removed the below text in the original manuscript. 

“On the global level, the numbers of old-old are expanding at the fastest rate, and, by 2050, are expected to make up 4.5 percent of total populations, compared to 1.9 percent in 2012 [9]. The age composition of Thailand’s population is on par with that of many developed countries; it is ranked as the third most rapidly ageing population in Asia, which now stands at about 13 million, accounting for 20% of the population [10]. By 2030, Thailand’s ageing population is expected to increase to 26.9% of the total population, equivalent to a quarter of the overall population [11]. This situation has resulted in a rise in chronic and degenerative diseases in countries worldwide. Indeed, in the 2015 global burden of disease, injuries, and risk factors study, neurological disorders were listed as the leading cause of disability globally. Amongst these, Parkinson’s disease (PD) has the fastest growing prevalence, disability rate, and mortality rate [12].”

3. This is a study of PD in old-old age, rather than of older onset PD. There are patients here with PD onset in middle age but long disease courses. The short and long disease duration comparison for the old-old group helps to highlight this.

Response: We would like to thank the reviewer for raising this important point in which additional analysis was undertaken to compare the old-old PD patients with short and long disease duration. Findings are shown in table 2 with supported text in the result sub-section: Old-old PD patients with short and long disease duration as following (page14-18). 

Table 2: Comparison of demographic and clinical characteristics for oldest-old PD patients with those disease duration <10 years versus those ≥10 years

 Disease duration p-value

 <10 (N = 30) ≥ 10 (N = 30) 

Demographic variables 

Current age, yrs, mean (�SD) 87.80 (�3.01) 88.70 (�3.196) 0.266

Age of PD onset, yrs, mean (�SD) 82.93 (�3.0) 72.53 (�5.21) <0.0001*

Disease duration, yrs, mean (�SD) 4.87 (2.69) 16.13 (�4.08) <0.0001*

Gender, male, N (%) 11 (36.7%) 13 (43.3%) 0.598

Annual outpatient visits, mean (�SD) 3.00 (�0.78) 2.60 (�0.67) 0.039 *

Motor symptoms 

Predominant subtype, N (%) 

TD 10 (33.3%) 3 (10%) 0.028

PIGD 20 (66.7%) 27 (90%) 

Motor severity 

UPDRS-III, mean (�SD) 30.13 (�13.19) 53.5 (�12.94) <0.0001*

H&Y, mean (�SD) 3.9 (�0.89) 4.58 (�0.72) 0.002*

Pi-H&Y(H&Y>3), N (%) 22 (73.3%) 27 (90%) 0.095

Non motor symptoms, N (%) 

NMSQuest total, mean (�SD) 10.57 (�2.64) 13.30 (�2.85) <0.0001*

Domain, N (%) 

Gastrointestinal 26 (86.7%) 28 (93.3%) 0.389

Urinary 17 (56.7%) 27 (90%) 0.004*

Sexual 10 (33.3%) 12 (40%) 0.592

CVS 10 (3.3%) 14 (46.7%) 0.292

Sleep/ fatigue 21 (70%) 22 (73.3%) 0.774

Apathy/attention/memory 23 (76.7%) 26 (86.7%) 0.317

Hallucination/ delusion 5 (16.7%) 13 (43.3%) 0.024*

Depression/ anxiety 6 (20%) 11 (36.7%) 0.152

Miscellaneous 20 (66.7%) 22 (73.3%) 0.573

TMSE, mean (�SD) 18.97(�6.98) 16 (�8.35) 0.141

Medications 

LEDD, mg/d, mean (�SD) 488.33(�267) 621.67(�387) 0.127

Motor complication, N (%) 

Dyskinesia 2 (6.7%) 8 (26.7%) 0.083

Wearing-off 11 (36.7%) 22 (73.3%) 0.004*

Disabilities 

S&E-ADL, mean (�SD) 55 (�22.1) 38 (�20.57) 0.003*

S&E-ADL ≥ 80%, N (%) 8 (26.7%) 3 (10%) 0.095

Milestones, N (%) 

Dementia 15 (50%) 19 (63.3%) 0.297

Recurrent falls in ambulatory patient (N=31), N (%) 6/ 22 (27.3%) 2/ 9 (22.2%) 0.771

Visual hallucination 8 (26.7%) 16 (53.3%) 0.035*

Nursing home placement 3(10%) 5(16.7%) 0.448

Wheelchair placement 12(40%) 20(66.7%) 0.038*

Hospitalization in past year, N (%) 9 (30%) 12 (40%) 0.417

Comorbidity, N (%) 

CVD 9 (30%) 12 (40%) 0.417

Musculoskeletal 15 (50%) 24 (80%) 0.015*

Hypertension 5 (16.7%) 7 (23.3%) 0.519

Diabetes 7 (23.3%) 6 (20%) 0.754

Cancer 3 (10%) 0 0.076

CCI, mean (�SD) 2.13 (�1.22) 2.26 (�1.20) 0.672

“Old-old PD patients with short and long disease duration: A general comparison

The 60 patients in the old-old groups were separated equally (30:30) into two subgroups, those with short disease duration (< 10 years) and those with long disease duration (≥ 10 years). Subgroups had the same mean age (p=0.266), but a mean disease duration of 4.87 and 16.13 years, respectively (p<0.0001). Gender ratio did not differ significantly (p=0.598). 

The long disease duration group showed higher motor scores on the UPDRS-III scale (p<0.0001) and average H&Y staging (p=0.002) than the short disease duration group; however, the frequency of the patients with H&Y stage 3 or higher was similar in both groups (p=0.095). Tremor as a predominant symptom was found less frequently in the longer disease duration group (p=0.028). The rate of outpatient visits in the long disease duration group (2.60 times per person per annum; pppa) was significantly lower than that of the short disease duration group (3.00 pppa). 

Nonmotor features

Patients with longer disease duration reported significantly higher mean NMSQuest scores than those with short disease duration (13.30 [� 2.85] vs. 10.57 [� 2.64], p<0.0001). More frequently reported problems by the patients with longer PD duration on the NMSQuest were in the domain of urinary problems (p=0.004), followed by hallucinations and delusions (p=0.024). There were no statistically significant differences in the remaining domains (gastrointestinal problems, p=0.389; sexual problems, p=0.592; cerebrovascular disease, p=0.292, cognitive deficits, p=0.317; sleep disturbances/fatigue, p=0.774; depression/anxiety, p=0.152; and miscellaneous, p=0.573).

In addition, there was no statistically significant difference between the two groups with respect to LEDD (p=0.127). Wearing-off was the most frequently reported motor complication in both groups, and, as expected, was associated with longer disease duration (p=0.004). Dyskinesia was much less frequent, and there was no significant difference between the two groups (p=0.083).

Comorbidities

There were no differences in comorbidity frequency or comorbidity severity (CCI, p=0.672) between the short and long disease duration groups, except for musculoskeletal disease (p=0.015).

Measurement of disease advancement

Concerns about disease milestones, visual hallucinations, and wheelchair-dependence were significantly more frequent in the long disease duration group (p=0.035 and p=0.038, respectively). Other advanced disease milestones were also more frequently reported in the long disease duration group, although these differences were not statistically significant.”

4. Both young-old and old-old patients are survivors of cohorts that were diagnosed with PD on average 10 years before. Can information be given about the size and outcome of the original cohorts? How many have died, or been lost to follow up? For old-old, this involves accounting for non-survivors who would have passed 85 years of age had they still been attending the clinic.

Response: It would have been interesting to explore this aspect since longitudinal study is indeed able to investigate the trends and relationship between the variables more thoroughly. Unfortunately, the limitations on the availability of data prior to electronic health record era prevent the secondary use when conducting research. We defined this point as a limitation, where it reads, “Finally, further potential limitation is the restricted collection of longitudinal data to a 6-year period according to hospital electronic health records. The temporal relationship variables and disease outcomes, thus, can be limited. We strongly suggest that longitudinal studies would enable a robust overall evidence base in a future research.” (page 27, line 531-535)

5. I can’t see criteria or references for subtyping of PD into postural instability-gait disturbance and tremor-dominant.

Response: As requested, references has been added (reference number 34, 35) 

Stebbins GT, Goetz CG, Burn DJ, Jankovic J, Khoo TK, Tilley BC. How to identify tremor dominant and postural instability/gait difficulty groups with the movement disorder society unified Parkinson's disease rating scale: comparison with the unified Parkinson's disease rating scale. Mov Disord. 2013;28(5):668-70.

Thenganatt MA, Jankovic J. Parkinson Disease Subtypes. JAMA Neurol.2014;71(4):499–504. doi:10.1001/jamaneurol.2013.6233

6. There is a tendency for readers to try to draw inferences about the longitudinal character of PD from cross-sectional or retrospective studies. This applies to quite a lot of PD research from the past. Point 4 above refers to one source of potential distortion. LIMITATIONS should discuss this issue.

Response: We agree that this is a potential limitation of the study. We have added the following sentences as a limitation (page 27, line 531-535) 

“Finally, further potential limitation is the restricted collection of longitudinal data to a 6-year period according to hospital electronic health records. The temporal relationship variables and disease outcomes, thus, can be limited. We strongly suggest that longitudinal studies would enable a robust overall evidence base in a future research.”

I would like to confirm that all authors have read the manuscript; the paper has not been previously published, and is not under simultaneous consideration by another journal. There is also no ghost writing by anyone not named on the author list.

There is no conflict of interest on all authors and we will take full responsibility for the data, the analyses and interpretation, and the conduct of the research. We had full access to all of the data; and that we had the right to publish any and all data, separate and apart from the attitudes of the sponsor. 

Thank you very much for consideration our manuscript for publication. Please let me know if there are any questions.

We are grateful to the editors and reviewers for the time and effort that they have put into helping us improve our manuscript.

Sincerely,

Roongroj Bhidayasiri

Corresponding author:

Roongroj Bhidayasiri, MD., FRCP., FRCPI.

Chulalongkorn Center of Excellence on Parkinson Disease and Related Disorders

Chulalongkorn University Hospital

1873 Rama 4 Road

Bangkok 10330

Thailand 

Tel: +662-256-4000 ext. 70701

Fax: +662-256-4630

Email address: rbh@chulapd.org

---

## [Decision Letter · Decision Letter 1]

6 Aug 2021

PONE-D-21-03548R1

Profiling the Most Elderly Parkinson’s Disease Patients: Does Age or Disease Duration Matter?

PLOS ONE

Dear Dr. Bhidayasiri,

Thank you for submitting your manuscript to PLOS ONE. After careful consideration, we feel that it has merit but does not fully meet PLOS ONE’s publication criteria as it currently stands. Therefore, we invite you to submit a revised version of the manuscript that addresses the points raised during the review process.

We look forward to receiving your revised manuscript.

Kind regards,

Karsten Witt

Academic Editor

PLOS ONE

Journal Requirements:

Additional Editor Comments (if provided):

Reviewers' comments:

Reviewer's Responses to Questions

**Comments to the Author**

1. If the authors have adequately addressed your comments raised in a previous round of review and you feel that this manuscript is now acceptable for publication, you may indicate that here to bypass the “Comments to the Author” section, enter your conflict of interest statement in the “Confidential to Editor” section, and submit your "Accept" recommendation.

Reviewer #1: (No Response)

Reviewer #2: (No Response)

2. Is the manuscript technically sound, and do the data support the conclusions?

Reviewer #1: Partly

Reviewer #2: Yes

3. Has the statistical analysis been performed appropriately and rigorously? 

Reviewer #1: Yes

Reviewer #2: Yes

4. Have the authors made all data underlying the findings in their manuscript fully available?

Reviewer #1: Yes

Reviewer #2: Yes

5. Is the manuscript presented in an intelligible fashion and written in standard English?

Reviewer #1: Yes

Reviewer #2: Yes

6. Review Comments to the Author

Reviewer #1: The authors have largely addressed my comments. However, the two groups of differing age with matched disease duration creates a major confound of age of onset in the interpretation of their results that should be addressed:

Major comments:

The selection of participants of two age groups with matched disease duration means that the two groups differ not only in age but also in age of onset. The authors’ interpretation of the differences in the two populations as being related to age is possible, but cannot be separated, then, from the age of onset and an equally valid interpretation would be that PD patients with later age of onset (rather than old-old) are more severe, have more non motor symptoms, etc than earlier age on onset (rather than young-old), as has been shown previously. This is fundamental to the analysis and interpretation of the results and should be addressed more than simply discussing in the limitations section. The authors could match for disease duration, and age of onset separately, e.g. The subgroup analysis that, in the old old group, the longer duration (hence younger age of onset) subgroup is more severe than the shorter duration subgroup is helpful; but would not address the alternate interpretation of the differences between the young-old and old-old group.

Minor comments:

Line 72: grammatical error needing clarification, are there 13 million elderly adults comprising 20% of the population?

Reviewer #2: Generally a satisfactory revision.

My Point 5: I think the subtyping criteria should be clearly stated in METHODS, not just referenced elsewhere.

7. PLOS authors have the option to publish the peer review history of their article (what does this mean?). If published, this will include your full peer review and any attached files.

Reviewer #1: No

Reviewer #2: No

---

## [Author Response · Author response to Decision Letter 1]

17 Oct 2021

17 September 2021

Dr. Karsten Witt

Academic Editor

PLOS ONE

Dear Dr. Witt,

Re: Manuscript # PONE-D-21-03548R1: Profiling the Most Elderly Parkinson’s Disease Patients: Does Age or Disease Duration Matter?

We are grateful to the editor and reviewers for their time and constructive comments on our manuscript. We have implemented their comments and suggestions and wish to submit a revised version of the manuscript for further consideration in the journal. Changes in the manuscript are highlighted in red in the revised version. Below, we also provide a point-by-point response explaining how we have addressed each of the editors or reviewers’ comments.

Reviewers' comments:

Reviewer #1: The authors have largely addressed my comments. However, the two groups of differing age with matched disease duration creates a major confound of age of onset in the interpretation of their results that should be addressed:

Response: Thank you for your comments. We have gone through your comments carefully and tried our best to address them one by one. We hope the manuscript has been improved accordingly.

Major comments:

The selection of participants of two age groups with matched disease duration means that the two groups differ not only in age but also in age of onset. The authors’ interpretation of the differences in the two populations as being related to age is possible, but cannot be separated, then, from the age of onset and an equally valid interpretation would be that PD patients with later age of onset (rather than old-old) are more severe, have more non motor symptoms, etc than earlier age on onset (rather than young-old), as has been shown previously. This is fundamental to the analysis and interpretation of the results and should be addressed more than simply discussing in the limitations section. The authors could match for disease duration, and age of onset separately, e.g. The subgroup analysis that, in the old old group, the longer duration (hence younger age of onset) subgroup is more severe than the shorter duration subgroup is helpful; but would not address the alternate interpretation of the differences between the young-old and old-old group.

Response: Thank you for raising this point. In this study, the primary objective is to fill the gap in the literature concerning the profile of old-old Parkinson’s (i.e., age ≥ 85 years old). The secondary objective is to demonstrate whether the driving force in each characteristic is worsening of the disease pathophysiology with increasing duration or age-related processes of the advanced age. 

However, aside from age and disease duration, as reviewer mentioned, age at onset may have several important clinical correlations. The authors used PubMed to search relevant literature using term “Parkinson’s disease” with additional search term “late onset”. These terms were restricted to the title of the article. Duplicates were removed and all articles published in English language (n=79) were review. Among these, 37 studies have specially analyzed characteristics in patients with Late Onset PD (LOPD) compare with Middle Onset PD (MOPD), and/or Young Onset PD (YOPD) matched for disease duration. A final explanation for observe differences in these studies could related to the old age of the old-age onset cohort, independent of onset age. To test this hypothesis, our subjects composed of old-old patients matched for age at the time of data collection but with different disease duration, i.e., the longer duration (MOPD) and shorter disease duration (LOPD). 

Furthermore, while there is still a lack of consensus in the age definition of early- and late-onset PD, several studies divided them into 3 distinct groups YOPD, MOPD and LOPD, and compared them with respect to the differences in clinical presentation, treatment, disease progression, biochemical profile, and nigrostriatal function. They found that the YOPD (≤ 50 years) group differs from MOPD and LOPD group in many aspects, whereas MOPD and LOPD patients are more similar (1, 2). This would imply that age of onset can contribute to the distinctive clinical-biochemical-pathological profile of YOPD from those in MOPD and LOPD, while the clinical impairment in LOPD may similarly be based in the make-up of the very old brain, including rate of nigrostriatal degeneration, reduced compensatory mechanisms, and frequency of comorbidities etc. This is also stated in previous studies (reference number 100, 101). To reflect this, and to avoid confusion due to the different pathophysiology bases of the YOPD, our studies specifically focused on MOPD and LOPD (>50 years). 

To address reviewer’s concern, although we reported the results using age at examination as a predictor variable (Table 1), we repeated the analysis using age at onset of PD as a comparator and details provided in the supplementary material (see S1 table). In general, findings were similar, and based on the make-up of the very old brain as mentioned earlier. To clarify this point, we also included the following statement in the methods, result and discussion of the manuscript read as follows:

Page 7: Line 160-164 

Moreover, to address the role of age at onset (AAO), we repeated the analysis in which patients were divided in 2 groups, middle-onset PD (MOPD) group included patients with AAO between 50 and 69, and the LOPD group included patients with AAO ≥70 years. This cut- off was guided by the conclusion of a cluster analysis from Post el al (3). Based on the available study subjects, we did not study those AAO of less than 50 years.

Page 19: Line 327 to Page 20: Line 353

MOPD and LOPD: a general comparison

Supplementary Data 1. shows details of demographic and clinical characteristics of the PD regarding to AOA. 97 patients (63%) belonged in the MOPD group and 55 (37%) in the LOPD group. Current age was 67.63 �6.05 and 88.07 �3.97 years in the MOPD and LOPD group, respectively (p<0.0001), whereas disease duration for both the groups was similar (10.93 vs. 9.04, p=0.054). Tremor or PIGD as the predominant motor symptom was not significantly different in MOPD and LOPD (p=0.544). UPDRS-III scores and H&Y stages were significantly higher in the LOPD patients compared to the MOPD patients (p<0.0001 and p<0.0001, respectively). 

Considering NMS, the mean NMSQuest score was significantly greater amongst the LOPD patients than the MOPD patients (p < 0.0001). More frequently reported NMS by MOPD patients were in the domain of sexual function (p=0.016), followed by depression and anxiety (p=0.024). While features of gastrointestinal problems (p<0.0001) and cognitive impairment (p<0.0001) were significantly more reported in the LOPD patients. There were no statistically significant differences in the remaining domains. 

When calculating the LEDD, the mean dosage in the MOPD patients was 559 mg and in the LOPD patients it was 871 mg (p<0.0001). Dyskinesia developed in 40% of the MOPD and 14.5% of the LOPD patients (p<0.001), while wearing-off rate did not differ between the groups (p=0.701). 

For disabilities and important clinical milestones, LOPD patients were more likely to have a greater disability, as measured by the S&E-ADL scale (p<0.0001) and tended to have higher milestone frequency. Specific milestones more common in LOPD than in MOPD patients were dementia (p<0.0001), wheelchair placement (p<0.0001), and hospitalisation in the past year (p=0.011). 

Page 28: Line 554 to Page 29: Line 566

This study aims to fill the gap in the literature concerning the profile of old-old Parkinson’s patients and to demonstrate whether the driving force in each characteristic is worsening of the disease pathophysiology with increasing duration or age-related processes of the advanced age. To address the role of AAO, we repeated our analysis using AAO of PD as a comparator, detailed in Supplementary data 1 and 2. In general, outcomes were similar to our other findings, as an inevitable consequence of variable-dependence around age, AAO and disease duration. Previous studies have suggested that AAO may contribute to the distinctive clinical-biochemical-pathological profile of patients with AAO ≤ 50 years (i.e., young-, early-, juvenile-onset PD; YOPD), while the different profile seen in patients with AAO > 50 years (middle- and late-onset PD) may be based on the make-up of the very old brain, including, rate of nigrostriatal degeneration, reduced compensatory mechanisms, and frequency of comorbidities (4, 5). To reflect this, and to avoid confusion due to different pathophysiology bases of YOPD, our studies specifically focused on those with AAO >50 years.

S1 table: Comparison of demographic and clinical characteristics of Middle Onset PD versus Late Onset PD patient (Submitted as supplementary data 1)

Furthermore, our study extended the analysis to the young-old with short and long disease duration, however, as to available study subjects, there were significant different in both age of onset and age at examination. See the supplementary material (S2 table) We also included the following statement in the methods and result section as following. 

Page 7: Line 160-164 

To investigate the influence of disease duration, patients of the old-old and young-old groups were also separated into two subgroups, according to duration of illness, into a ‘less than 10 years’ group, called the short duration group, and an ‘equal to or more than 10 years’ group, called the long duration group, based on a commonly used cut-off point from previously published studies (6, 7).

Page 20: Line 356-358

Our study extended these comparisons to the young-old patients with short and long disease duration. However, as to the available subjects, there was a significant different in age at examination (Supplementary Data 2).

S2 table: Comparison of demographic and clinical characteristics for younger-old PD patients with those disease duration <10 years versus those ≥10 years (Submitted as supplementary data 2)

Minor comments:

Line 72: grammatical error needing clarification, are there 13 million elderly adults comprising 20% of the population?

Response: 

- We have corrected this grammatical error in line 72. 

- The data source was the “Policy Mapping on Ageing in Asia and the Pacific Analytical Report”. We already provided the following citations to support this statement (Reference number 4)

4. HelpAge International. 2015. Policy Mapping on Ageing in Asia and the Pacific Analytical Report. Chiang Mai: HelpAge International East Asia/Pacific Regional Office.

Reviewer #2: Generally a satisfactory revision.

My Point 5: I think the subtyping criteria should be clearly stated in METHODS, not just referenced elsewhere

Response: Thank you for your nice reminder. We already provided an explanation in METHODS, Subsection Measurement of clinical variables as following (see page 8, line 174-179). 

“PD motor subtypes were identified following the original classification methods into two subtypes: (1) tremor- dominant PD and (2) postural instability and gait difficulty (PIGD), using the UPDRS, an average global tremor score and a mean score for the complex of PIGD were determined, and patients were assigned to a tremor group and a PIGD group based on the ratio of these scores (8, 9).”

I would like to confirm that all authors have read the manuscript; the paper has not been previously published, and is not under simultaneous consideration by another journal. There is also no ghost writing by anyone not named on the author list.

There is no conflict of interest on all authors and we will take full responsibility for the data, the analyses and interpretation, and the conduct of the research. We had full access to all of the data; and that we had the right to publish any and all data, separate and apart from the attitudes of the sponsor. 

Thank you very much for consideration our manuscript for publication. Please let me know if there are any questions.

We are grateful to the editors and reviewers for the time and effort that they have put into helping us improve our manuscript.

Sincerely,

Corresponding author:

Roongroj Bhidayasiri, MD., FRCP., FRCPI.

Chulalongkorn Center of Excellence on Parkinson Disease and Related Disorders

Chulalongkorn University Hospital

1873 Rama 4 Road

Bangkok 10330

Thailand 

Tel: +662-256-4000 ext. 70701

Fax: +662-256-4630

Email address: rbh@chulapd.org

References

1. Mehanna R, Moore S, Hou JG, Sarwar AI, Lai EC. Comparing clinical features of young onset, middle onset and late onset Parkinson's disease. Parkinsonism Relat Disord. 2014;20(5):530-4.

2. Schirinzi T, Di Lazzaro G, Sancesario GM, Summa S, Petrucci S, Colona VL, et al. Young-onset and late-onset Parkinson's disease exhibit a different profile of fluid biomarkers and clinical features. Neurobiol Aging. 2020;90:119-24.

3. Post B, Speelman JD, de Haan RJ. Clinical heterogeneity in newly diagnosed Parkinson's disease. J Neurol. 2008;255(5):716-22.

4. Diederich NJ, Moore CG, Leurgans SE, Chmura TA, Goetz CG. Parkinson Disease With Old-Age Onset: A Comparative Study With Subjects With Middle-Age Onset. Archives of Neurology. 2003;60(4):529-33.

5. Levy G, Louis ED, Cote L, Perez M, Mejia-Santana H, Andrews H, et al. Contribution of aging to the severity of different motor signs in Parkinson disease. Arch Neurol. 2005;62(3):467-72.

6. Hely MA, Morris JGL, Traficante R, Reid WGJ, O’Sullivan DJ, Williamson PM. The Sydney multicentre study of Parkinson’s disease: progression and mortality at 10 years. Journal of Neurology, Neurosurgery & Psychiatry. 1999;67(3):300.

7. Hassan A, Wu SS, Schmidt P, Malaty IA, Dai YF, Miyasaki JM, et al. What are the issues facing Parkinson's disease patients at ten years of disease and beyond? Data from the NPF-QII study. Parkinsonism Relat Disord. 2012;18 Suppl 3:S10-4.

8. Stebbins GT, Goetz CG, Burn DJ, Jankovic J, Khoo TK, Tilley BC. How to identify tremor dominant and postural instability/gait difficulty groups with the movement disorder society unified Parkinson's disease rating scale: comparison with the unified Parkinson's disease rating scale. Mov Disord. 2013;28(5):668-70.

9. Thenganatt MA, Jankovic J. Parkinson Disease Subtypes. JAMA Neurology. 2014;71(4):499-504.

---

## [Decision Letter · Decision Letter 2]

1 Dec 2021

Profiling the Most Elderly Parkinson’s Disease Patients: Does Age or Disease Duration Matter?

PONE-D-21-03548R2

Dear Dr. Bhidayasiri,

We’re pleased to inform you that your manuscript has been judged scientifically suitable for publication and will be formally accepted for publication once it meets all outstanding technical requirements.

Kind regards,

Karsten Witt

Academic Editor

PLOS ONE

Additional Editor Comments (optional):

Reviewers' comments:

Reviewer's Responses to Questions

**Comments to the Author**

1. If the authors have adequately addressed your comments raised in a previous round of review and you feel that this manuscript is now acceptable for publication, you may indicate that here to bypass the “Comments to the Author” section, enter your conflict of interest statement in the “Confidential to Editor” section, and submit your "Accept" recommendation.

Reviewer #1: All comments have been addressed

Reviewer #2: All comments have been addressed

2. Is the manuscript technically sound, and do the data support the conclusions?

Reviewer #1: Yes

Reviewer #2: (No Response)

3. Has the statistical analysis been performed appropriately and rigorously? 

Reviewer #1: Yes

Reviewer #2: (No Response)

4. Have the authors made all data underlying the findings in their manuscript fully available?

Reviewer #1: Yes

Reviewer #2: (No Response)

5. Is the manuscript presented in an intelligible fashion and written in standard English?

Reviewer #1: Yes

Reviewer #2: (No Response)

6. Review Comments to the Author

Reviewer #1: (No Response)

Reviewer #2: (No Response)

7. PLOS authors have the option to publish the peer review history of their article (what does this mean?). If published, this will include your full peer review and any attached files.

Reviewer #1: No

Reviewer #2: No

---

## [Editor Report · Acceptance letter]

6 Dec 2021

PONE-D-21-03548R2 

Profiling the Most Elderly Parkinson’s Disease Patients: Does Age or Disease Duration Matter? 

Dear Dr. Bhidayasiri:

I'm pleased to inform you that your manuscript has been deemed suitable for publication in PLOS ONE. Congratulations! Your manuscript is now with our production department. 

Kind regards, 

on behalf of

Dr. Karsten Witt 

Academic Editor

PLOS ONE